# Purines enrich root-associated *Pseudomonas* and improve wild soybean growth under salt stress

Yanfen Zheng [1,4], Xuwen Cao[2,4], Yanan Zhou[1,3], Siqi Ma[1], Youqiang Wang[1], Zhe Li[1], Donglin Zhao[1], Yanzhe Yang[1], Han Zhang[1], Chen Meng[1], Zhihong Xie[3], Xiaona Sui[1], Kangwen Xu[1], Yiqiang Li[1] & Cheng-Sheng Zhang [1] ✉

The root-associated microbiota plays an important role in the response to environmental stress. However, the underlying mechanisms controlling the interaction between salt-stressed plants and microbiota are poorly understood. Here, by focusing on a salt-tolerant plant wild soybean (*Glycine soja*), we demonstrate that highly conserved microbes dominated by *Pseudomonas* are enriched in the root and rhizosphere microbiota of salt-stressed plant. Two corresponding *Pseudomonas* isolates are confirmed to enhance the salt tolerance of wild soybean. Shotgun metagenomic and metatranscriptomic sequencing reveal that motility-associated genes, mainly chemotaxis and flagellar assembly, are significantly enriched and expressed in salt-treated samples. We further find that roots of salt stressed plants secreted purines, especially xanthine, which induce motility of the *Pseudomonas* isolates. Moreover, exogenous application for xanthine to non-stressed plants results in *Pseudomonas* enrichment, reproducing the microbiota shift in salt-stressed root. Finally, *Pseudomonas* mutant analysis shows that the motility related gene *cheW* is required for chemotaxis toward xanthine and for enhancing plant salt tolerance. Our study proposes that wild soybean recruits beneficial *Pseudomonas* species by exudating key metabolites (i.e., purine) against salt stress.

Salinity is one of the most frequently encountered but least studied environmental stresses, and has greatly reduced agricultural productivity. Salt-tolerant plants have evolved many strategies to resist salt stress, including adjusting cellular osmotic pressure by biosynthesis of osmoprotectants[1], secreting salt out of plants via glands or trichomes[2], and maintaining cellular redox equilibrium[3]. Recently, plant-associated microbes have been considered as key members in enhancing stress tolerance by improving plant growth[4]. Various beneficial microbes reside in root-associated environments of halophytes[5–8], including *Pseudomonas*, *Bacillus*, and *Arthrobacter*[9].

They can alleviate the adverse effects of salt stress on plant growth through various physiological regulatory processes, such as maintaining ion homeostasis and altering endogenous hormone status[10]. Studies on the genetic and molecular mechanisms underlying the plant-microbe interactions may provide insights into harnessing the rhizosphere microbiome to promote plant performance and reduce salinity-induced crop losses.

Microbiota refers to the microbial community in a particular environment, while the microbiome comprises microbiota and their structural elements (such as nucleic acids and proteins) and

[1]Marine Agriculture Research Center, Tobacco Research Institute of Chinese Academy of Agricultural Sciences, Qingdao 266101, China. [2]Institute of Marine Science and Technology, Shandong University, Qingdao 266200, China. [3]National Engineering Laboratory for Efficient Utilization of Soil and Fertilizer Resources, College of Resources and Environment of Shandong Agricultural University, Taian 271018, China. [4]These authors contributed equally: Yanfen Zheng, Xuwen Cao. ✉e-mail: zhangchengsheng@caas.cn

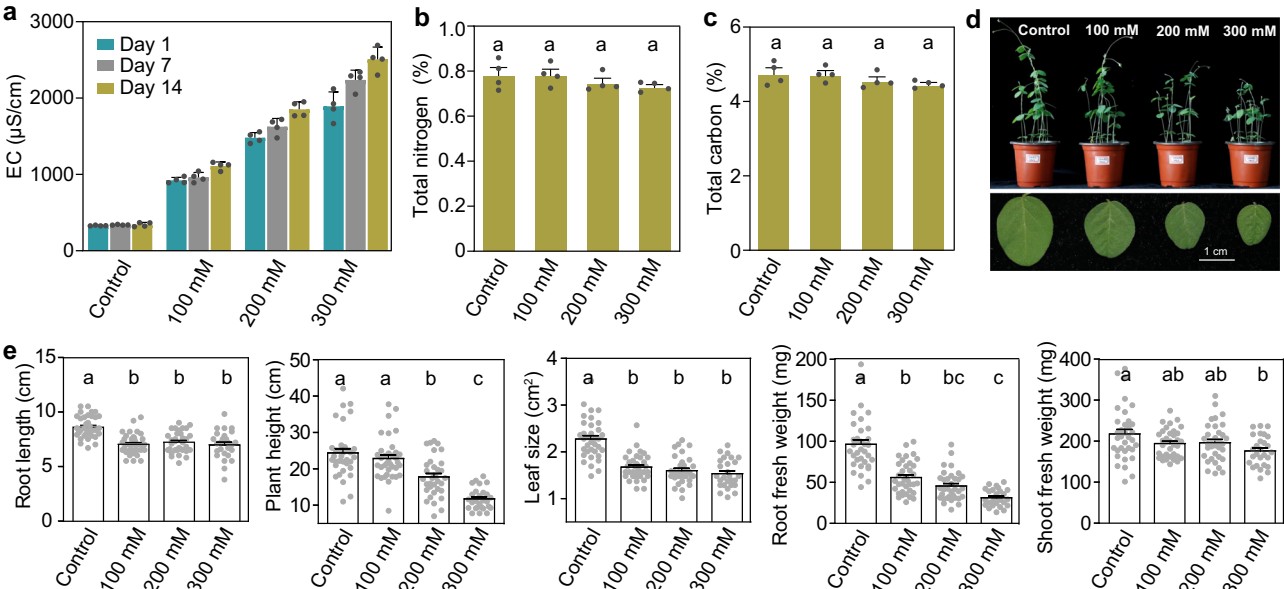

**Fig. 1 | The soil physicochemical properties and wild soybean growth affected by salt stress. a** Soil EC of control (0 mM NaCl) and salt treatments (100, 200 and 300 mM NaCl) after 1, 7 and 14 days of salt stress (n = 4 biologically independent samples). Abbreviation: EC, electrical conductivity. **b** Soil total nitrogen of control and salt treatments after 14 days of salt stress (n = 4 biologically independent samples). **c** Soil total carbon of control and salt treatments after 14 days of salt stress (n = 4 biologically independent samples). **d** The growth of wild soybean under control and different salt treatments. **e** Root length, plant height, leaf size, root weight and shoot weight of wild soybean in control and different salt treatments. The number of samples per treatment in (**e**) is as follows: control (n = 35), 100 mM (n = 36), 200 mM (n = 34), and 300 mM (n = 27). All data in this figure are mean ± SEM. Different letters above the bars in figures (**b, c, e**) indicate a significant difference at P < 0.05 (one-way ANOVA with correction by Tukey's HSD test, P-values are shown in source data). Source data are provided as a Source Data file.

metabolites[11]. Abiotic stresses can cause the root-associated microbiome shift and induce the recruitment of stress-alleviating microbes[12–16]. For example, drought stress leads to a conserved shift in microbiota composition with enrichment of the genus *Streptomyces*, which shows growth-promoting properties under drought condition[14,17], and the underlying causes of this microbial enrichment have also been speculated[18]. Many studies have shown that salt stress can affect the composition of root-associated microbiota in barley[19], wheat[20], rice[21], caliph medic[22], date palms[23], and plants belonging to the family Curcurbitaceae[15]. However, these studies have mainly focused on microbiota shifts among different plant genotypes and species (such as salt-tolerant and salt-sensitive plants), and identified a few salt-responsive taxa. Thus, a detailed understanding of the response of the root microbiome and the underlying genetic mechanisms driving these microbial changes under salt stress are urgently needed.

Roots are frequently exposed to various abiotic stresses, and they respond to these stresses by excreting various metabolites that are important drivers in modulating the assembly of the root microbiome to alleviate the encountered stress[24–27]. For instance, flavones (e.g., apigenin and luteolin) were produced under nitrogen deprivation, which mediated the enrichment of the family Oxalobacteraceae[28]. Other abiotic stress conditions, such as iron and phosphate deficiencies, induced increased production of coumarins[29] and strigolactones (or indole glucosinolates)[30,31] respectively, possibly modulating the root microbiota. Previous studies observed that salt-stressed plants secreted larger amounts of metabolites, including phenolic compounds, amino acids, organic acids, and sugars, than non-stressed plants[32,33]. However, their interactions with the altered root-associated microbiome under salt stress have not been investigated.

Here, we hypothesized that under salt stress, plants recruit specific microbes to enhance salt tolerance, and this recruitment is mediated by associated key root metabolites. To test this hypothesis, we focused on wild soybean (*Glycine soja*), a salt-tolerant plant naturally living in the saline soils. We examined the shift in the composition

and function of root-associated microbiota under salt stress condition using 16S rRNA gene amplicon, and metagenomic and metatranscriptomic sequencing. The stress-responsive microbiota was identified, and their role in promoting plant growth were assessed. Finally, the underlying causes of salt-stressed community changes were elucidated based on the genetic properties of the enriched microbes and root metabolites in wild soybean.

## Results
### Wild soybean growth is affected by salt stress
Wild soybean seeds were grown in agricultural soil, and three NaCl concentrations (100, 200 and 300 mM) were used to stress wild soybean plants. To determine the salt stress levels, we measured the soil electrical conductivity (EC), which reflects the available ion content in the soil. Expectedly, the increasing soil EC was observed in all three salt-treated groups (Fig. 1a). Additionally, no discernible differences in the content of soil total carbon (TC) and total nitrogen (TN) were observed between the salt-treated and control groups (Fig. 1b, c), suggesting that short-term salt stress did not cause obvious fluctuations in the soil chemical properties. However, salt stress severely affected plant performance (Fig. 1d), resulting in the reduction of the root length, plant height, leaf size, and root and shoot fresh weights, especially in wild soybean treated with 300 mM NaCl (Fig. 1e). Overall, these results indicate that salt stress increases soil EC and affects the growth of wild soybean.

### Salt stress induces distinct responses in bacterial diversity and dominant taxa among compartments
To examine how salt stress affects bacterial recruitment, we monitored the bacterial community composition in the root, rhizosphere soil, and bulk soil using Illumina MiSeq sequencing of the V5–V7 region of the 16S rRNA gene across three time points (1, 7, and 14 days after salt stress). Comparison of the control and salt treatments revealed no discernible differences in Shannon's diversity within the root at early salt stress (Day 1), whereas all salt treatments (except for the 300 mM

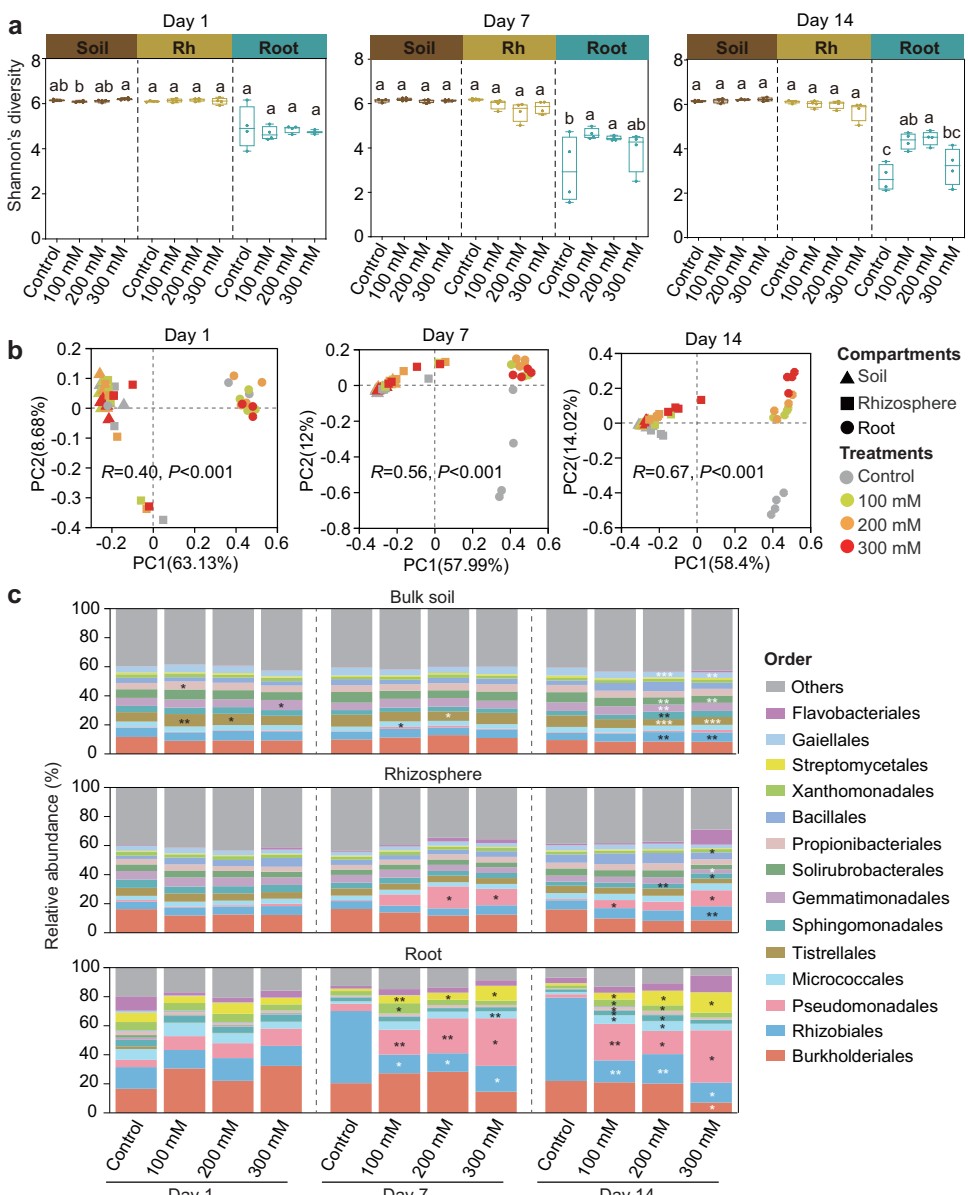

**Fig. 2 | The impacts of salt stress on bacterial diversity and composition of wild soybean based on 16S rRNA gene amplicon data. a** The impacts of salt stress on bacterial diversity (Shannon index) of bulk soil, rhizosphere soil and root after 1, 7 and 14 days of salt stress ($n = 4$ biologically independent samples, mean ± SEM). Different letters above the boxes of each compartment indicate a significant difference at $P < 0.05$ (one-way ANOVA with correction by Tukey's HSD test, $P$-values are shown in source data). Abbreviation: Rh, rhizosphere. **b** PCoA ordination of the Bray-Curtis dissimilarity matrix (OTU level) for control and salt treatments across bulk soil, rhizosphere soil and root after 1, 7 and 14 days of salt stress. Statistical analysis was performed using ANOSIM (analysis of similarities). **c** Percent relative abundance of the top 15 most abundant orders for bulk soil, rhizosphere soil, and root across control and different levels of salt stress. Different significance levels between control and each salt treatment are marked with asterisks (*$P < 0.05$, **$P < 0.01$ and ***$P < 0.001$, two-sided Student's $t$-test, $P$-values are shown in source data). Low abundance orders (< 1.5%) with significant difference are not marked with asterisks. White asterisks indicate the relative abundance of this order is lower in salt treatments than that in control group. Dark asterisks indicate the relative abundance of this order is higher in salt treatments than that in control group. Source data are provided as a Source Data file.

NaCl treatment) led to a significantly increased level of Shannon's diversity at Day 7 and Day 14. The decreased Shannon's diversity in the 300 mM NaCl treatment in contrast to other salt-treated samples was possibly because the high salt concentration greatly affected the growth of wild soybean, limiting the recruitment of some microbial species. Furthermore, unchanged levels of Shannon's diversity in both the rhizosphere soil and bulk soil were observed between the control and salt treatments at all time points (Fig. 2a). Permutational multivariate analysis of variance (PERMANOVA) analysis based on Bray Curtis distances suggested that the compartment was the strongest deterministic force controlling microbiota assembly, followed by time point and treatment (Supplementary Table 1). The root community (analysis of similarity [ANOSIM]; $R = 0.76$, $P < 0.01$; at Day 14) showed greater compositional changes after salt stress than the other two compartments (ANOSIM; rhizosphere soil: $R = 0.41$, $P = 0.002$; bulk soil: $R = 0.27$, $P = 0.023$; at Day 14), which were also revealed by a larger separation between the control and salt treatments in the root samples (Fig. 2b and Supplementary Fig. 1). These results indicate that salt stress exhibits a greater effect on root communities.

To determine the effect of salt on root-associated microbiota, we analyzed the phylum- and order-level compositional profiles of the control and salt-treated samples. After 1–2 weeks (Day 7 and Day 14) of

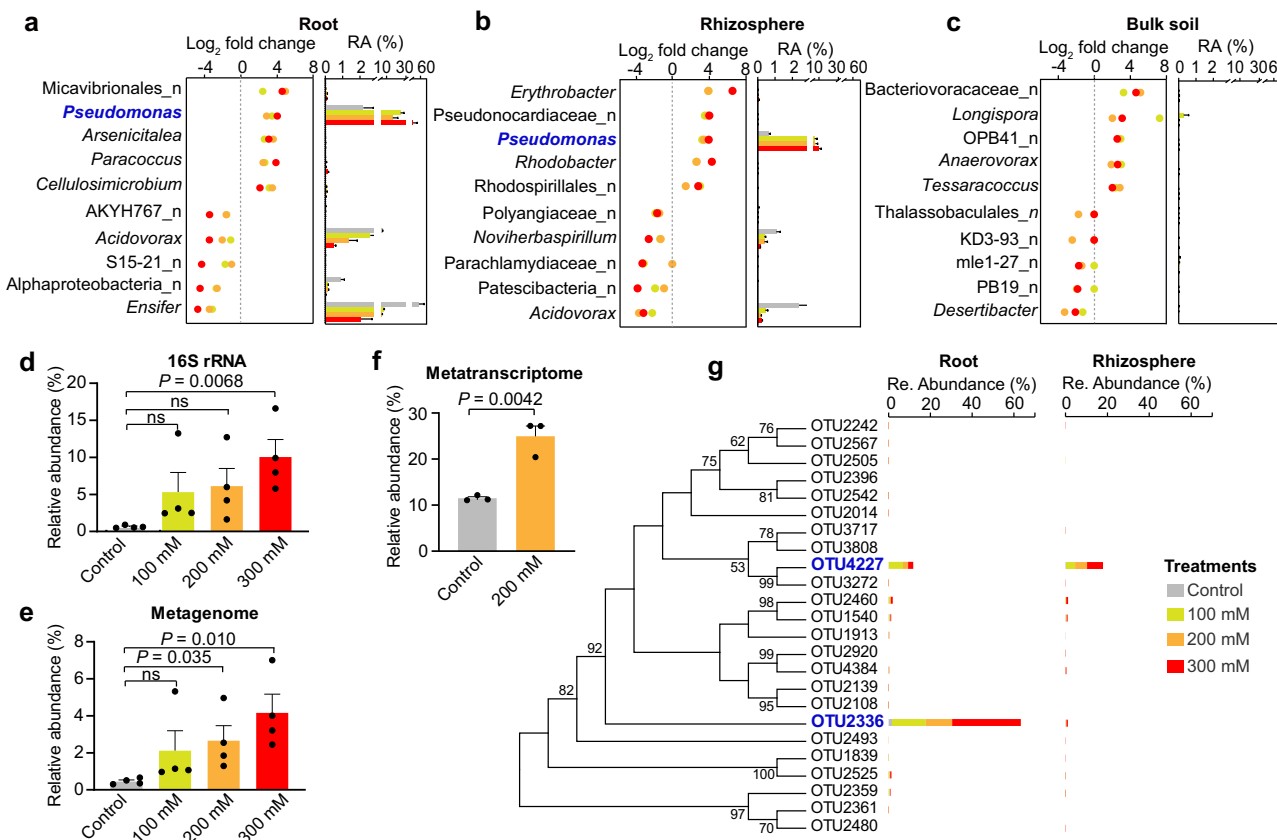

**Fig. 3 | *Pseudomonas* enriched in salt-treated root and rhizosphere samples.**
**a–c** The enriched or depleted genera by different levels of salt stress across root (**a**), rhizosphere (**b**) and bulk soil (**c**) are visualized with dot plot (left). The relative abundance of each genus in different treatments is shown with bar plot (right). Blue word indicates the most abundant taxon among all enriched genera. The "_n" prefix in figures (**a**–**c**) indicate the taxonomic assignment was unknown at the genus level. **d** The relative abundance of *Pseudomonas* in the rhizosphere soil based on 16S rRNA gene amplicon data. **e** The relative abundance of *Pseudomonas* in the rhizosphere soil based on metagenomic data. **f** The relative abundance of *Pseudomonas*

in the rhizosphere soil based on metatranscriptomic data. Values are means ± SEM for (**d**) (*n* = 4 biologically independent samples), (**e**) (*n* = 4 biologically independent samples), and (**f**) (*n* = 3 biologically independent samples). Significance between control and each treatment are determined by two-sided Student's *t*-test.
**g** Neighbor-Joining tree of all *Pseudomonas* OTUs and their relative abundance in control and salt treated samples across root and rhizosphere compartments. The predominant OTUs are labled in blue words. Source data are provided as a Source Data file.

salt stress, an increase in the relative abundances of Gammaproteo-bacteria and Actinobacteriota was observed in the root community (Supplementary Fig. 2). No discernible taxonomic changes were found between the control and salt treatments in either the rhizosphere soil or bulk soil. At the order level, the relative abundances of Pseudo-monadales and Streptomycetales increased significantly in all salt-treated root samples at Day 7 and Day 14, compared with those in the control group (Fig. 2c). An increase in Pseudomonadales abundance was also observed in the rhizosphere soil communities but not in the bulk soil (Fig. 2c).

Next, we analyzed the genus-level composition to identify the genera recruited by salt-stressed wild soybean. After 14 days of salt stress, 130 of 598 genera (21.74%) in the root, 56 of 776 genera (7.22%) in the rhizosphere soil, and 38 of 779 genera (4.88%) in the bulk soil showed significant enrichment or depletion in at least one salt treat-ment (Supplementary Fig. 3 and Supplementary Data 1). The enriched genera with a high relative abundance (cumulative relative abundance of > 20%) in the salt-treated root and rhizosphere samples were only distributed in either Gammaproteobacteria or Actinobacteriota (Sup-plementary Fig. 3a, b). To determine if the enrichment in Actino-bacteriota and Gammaproteobacteria was caused by an increase in the total bacterial absolute abundance, we quantified the total bacterial abundance in the rhizosphere soil and root samples using qPCR. Compared with the control group, the total bacterial absolute

abundances showed some fluctuations among the three salt treat-ments, but the changes were not significant in either the rhizosphere soil or root samples (Supplementary Fig. 4), suggesting that the enrichment of Actinobacteriota and Gammaproteobacteria was not the result of an increase in total bacterial absolute abundances but a decrease in the abundances of other communities. Overall, these results indicate that wild soybean root, rhizosphere soil, and bulk soil microbiota respond differently to salt stress across the taxonomic profile.

## *Pseudomonas* abundance increases dramatically in salt-treated roots and rhizosphere soils

Among the top 10 most changed genera, *Pseudomonas* (enriched in salt treatments) showed the highest abundance (cumulative relative abundance >22.17%) in both the root and rhizosphere samples (Fig. 3a, b), whereas all genera changed in the bulk soil were found in low abundance (all < 0.1%) (Fig. 3c). *Pseudomonas* was enriched in all three salt-treated roots, and its relative abundance increased from 2.19% in the control group to 25.00%, 16.58%, and 37.42% in the 100, 200, and 300 mM NaCl treatments, respectively (Fig. 3a). Within the rhizospheres, *Pseudomonas* was also highly enriched in salt treatments by up to 16-folds than that in the control group (Fig. 3b, d). Consistent with the results of the 16S rRNA gene amplicon analysis, metagenomic DNA- and RNA-based analyses of the rhizosphere microbiota showed

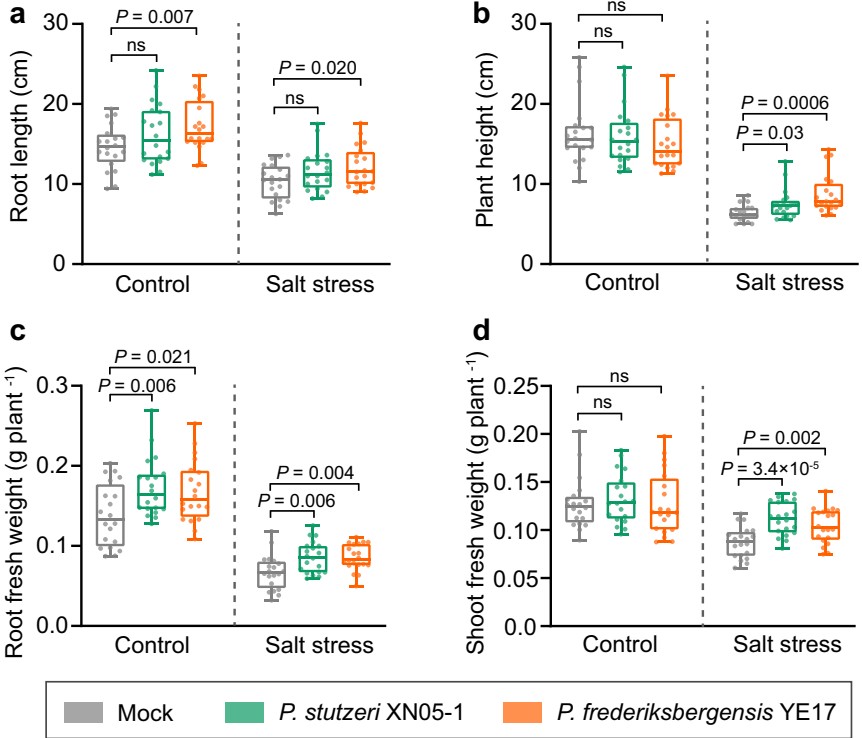

**Fig. 4 | Plant phenotypes after *Pseudomonas* inoculation.** Root length (**a**), plant height (**b**), root fresh weight (**c**) and shoot fresh weight (**d**) of wild soybean inoculated without (Mock) or with strains XN05-1 and YE17 under control and salt stress condition. Tops and bottoms of boxes represent 25th and 75th percentiles, respectively. Horizontal bars within boxes denote medians, and the upper and lower whiskers represent the range of non-outlier data values. All plots are mean ± SEM ($n = 20$ plants). *P*-values are calculated by two-sided Student's *t*-test. ns, not significant. Source data are provided as a Source Data file.

that *Pseudomonas* was not only the most abundant member (Fig. 3e) but also was active with a relative abundance of transcripts of 21.94% in the salt-treated sample and 11.49% in the control (Fig. 3f). To evaluate if the feature of *Pseudomonas* enrichment was reproducible in another soil and plant, independent salt experiments were performed using domesticated soybean plant (*G. max*) and another Shandong soil (collected from ~ 400 km away from the soil used in this study; pH: 8.8 ± 0.2, EC: 175.3 ± 30.2 µs/cm; Supplementary Fig. 5). The results showed that the relative abundance of *Pseudomonas* was higher under salt stress than that in control in all cases (Supplementary Fig. 6), indicating that salt-induced *Pseudomonas* enrichment was conserved across different microbiome backgrounds. Nevertheless, studies on more soil types and plants are required to validate if salt-induced *Pseudomonas* enrichment is widespread.

Since *Pseudomonas* was highly prevalent in the salt-treated root and rhizosphere communities, we further investigated the compositional trends of each individual OTU within this genus. A total of 24 OTUs were assigned as *Pseudomonas*, of which, OTU4227 was the most abundant taxon in the rhizosphere soil (4.40–7.54% in salt treatments). However, OTU4227 was less predominant (2.37–6.80%) in the root, whereas OTU2336 was the notably abundant taxon, reaching a relative abundance range of 12.57–32.81% in salt treatments (Fig. 3g). Thus, OTU4227 was enriched by salt stress in both the root and rhizosphere soil, whereas OTU2336 exhibited an ecological niche preference. Collectively, these results indicate that salt-induced *Pseudomonas* enrichment occurs in the root and/or rhizosphere soil.

**Two *Pseudomonas* isolates rescued wild soybean growth under salt stress**

To assess if this highly enriched *Pseudomonas* taxon contributed to wild soybean growth under salt stress, we performed an isolation campaign from the roots of salt-stressed plants. A total of 34

*Pseudomonas* isolates were obtained (Supplementary Table 2), and their sequences were compared with the two most prevalent sequences, OTU2336 and OTU4227. None of the isolates obtained in this study shared > 98% similarity with OTU4227 (Supplementary Table 2). However, one isolate, namely *Pseudomonas frederiksbergensis* YE17, showed 99.21% 16S rRNA gene similarity with OTU2336 (Supplementary Fig. 7 and Supplementary Table 2). Additionally, metagenomic data revealed that *Pseudomonas stutzeri* was the most abundant *Pseudomonas* species in the rhizosphere soil treated with 300 mM NaCl (Supplementary Fig. 8a), and this species was available in our *Pseudomonas* isolation campaign (*P. stutzeri* XN05-1; Supplementary Table 2). The number of metagenomic reads mapped to the XN05-1 genome was higher in the salt treatments than in the control (Supplementary Fig. 8b), suggesting that this strain represents the actual DNA sequences enriched in salt treatments. Thus, two strains, *P. stutzeri* XN05-1 and *P. frederiksbergensis* YE17, were used as representatives of salt-enriched *Pseudomonas* to test their effects on wild soybean growth in a controlled inoculation experiment using sterile soil.

Wild soybean seeds were grown for 14 days in sterile soil, followed by a 10-day period of salt stress. Strains XN05-1 and YE17 were inoculated into the soil before salt stress. The root and shoot growth parameters of wild soybean were measured. We observed that under non-salt stress condition, only strain YE17 increased root growth but not shoot growth (Fig. 4). However, under salt stress, *Pseudomonas* isolates XN05-1 and YE17 significantly improved root and shoot growths (including length and fresh weight) of wild soybean (Fig. 4). This indicates that the plant growth enhancement induced by these strains was primarily specific to salt stress condition. To confirm that *Pseudomonas* isolates colonized the roots of wild soybean, we inoculated these two strains in sterile soil in equal amounts to quantify the levels of *Pseudomonas* colonization under salt and non-salt conditions using qPCR with specific primers. The results showed that the absolute

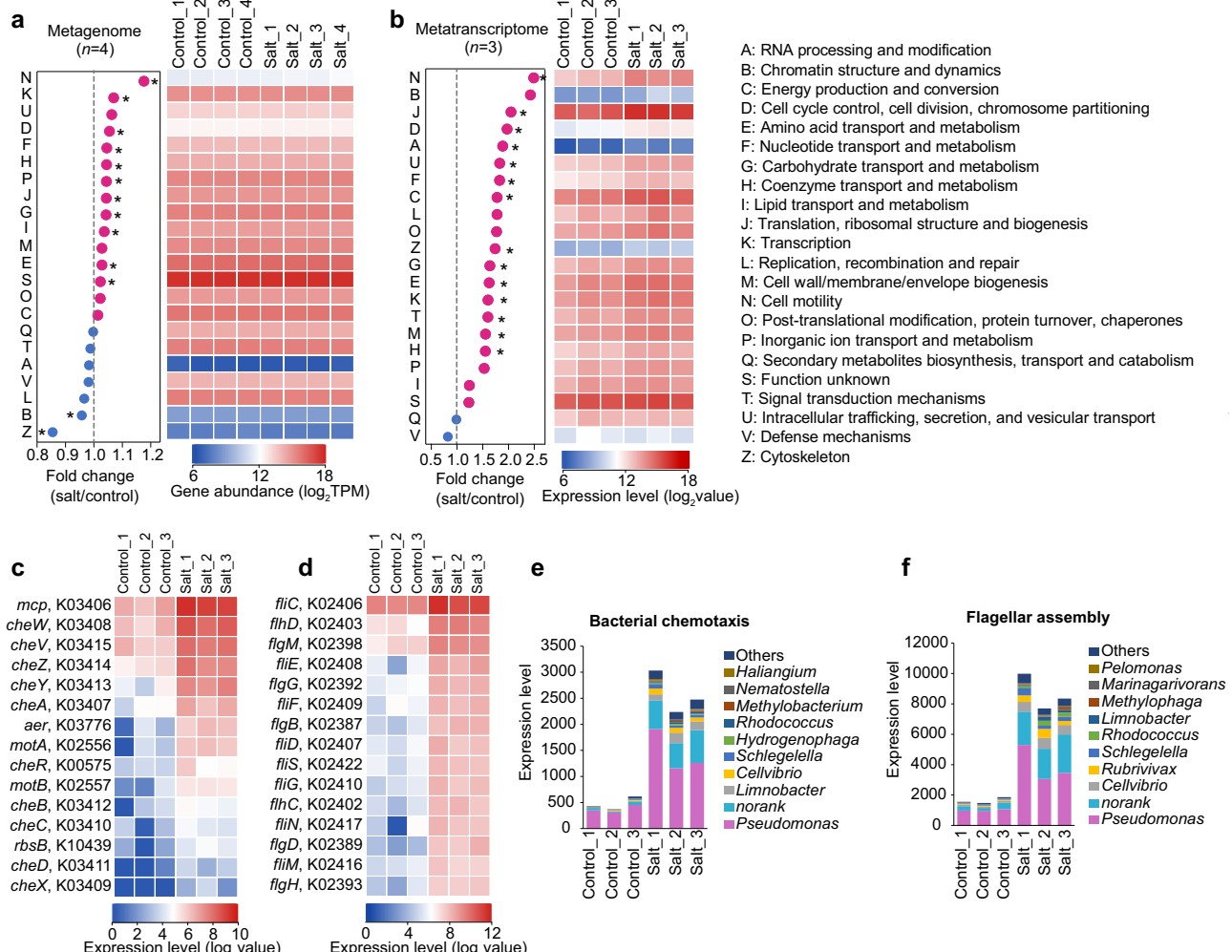

**Fig. 5 | The salt induced functional shift of rhizosphere microbiota. a** The fold change (left) and gene abundance (right) of COG categories in the control and salt treated rhizosphere soil determined by metagenomic analysis. **b** The fold change (left) and expression level (right) of differentially expressed genes (DEGs) in the control and salt treated rhizosphere soil determined by metatranscriptomic analysis. Low abundance categories in all samples, i.e., extracellular structures (COG W)

and nuclear structure (COG Y), were not shown. *P*-values were calculated by two-sided Student's *t*-test (*adjusted *P* < 0.05 by Benjamini and Hochberg method). **c, d** The top 15 highest expression levels of DEGs annotated as bacterial chemotaxis (**c**) and flagellar assembly subcategories (**d**). **e, f** The taxonomic annotation of bacterial chemotaxis (**e**) and flagellar assembly DEGs (**f**). Source data are provided as a Source Data file.

abundance of *Pseudomonas* was higher under salt stress compared with non-salt condition (Supplementary Fig. 9), which is consistent with the finding that salt stress enriches *Pseudomonas*.

## Cell motility genes and transcripts increased in salt-treated rhizosphere

To explore if the salt-induced community shifts in the rhizosphere were correlated with changes in microbial functions, shotgun metagenome sequencing of the rhizosphere microbiota was performed, and the relative abundances of key functional pathways were analyzed. According to the metagenomic data annotated against the COG database, we observed significant differences between the control and three salt treatments (ANOSIM: control versus 100 mM NaCl treatment: $R = 0.135$, $P = 0.157$; control versus 200 mM NaCl treatment: $R = 0.458$, $P = 0.034$; and control versus 300 mM NaCl treatment: $R = 0.646$, $P = 0.034$). As the samples treated with 300 mM NaCl showed a larger difference from the control than the other two salt treatments, this group was used to for further refined analyses. COG analysis revealed that cell motility (COG category: N), transcription (COG category: K), and several biogenesis-, transport-, and metabolism-related processes were markedly enriched in salt-treated

rhizosphere soil (Fig. 5a). Notably, the fold change of genes affiliated to cell motility was highest among all COG categories.

To gain a more in-depth understanding of the functions expressed under salt stress, we performed an independent salt stress experiment and extracted high-quality total RNA from the rhizosphere soil of the control and salt-treated plants. After illumina sequencing, we obtained a total of 189,319 genes in our dataset, of which, 96,803 genes were identified as the differentially expressed genes (DEGs). COG function analysis revealed that DEGs associated with cell motility (COG category: N) and translation, ribosomal structure, and biogenesis (COG category: J) were 2.5- and 2.0-folds enriched in salt treatment, respectively (Fig. 5b). As cell motility genes were highly transcribed in the microbial populations of the salt-treated samples, a finer-resolution analysis of this category was performed. Bacterial chemotaxis and flagellar assembly are two subcategories of cell motility and both were significantly enriched in the salt treatment group (Supplementary Fig. 10). We found that 395 DEGs were assigned to bacterial chemotaxis (329 upregulated and 66 downregulated genes) and 729 DEGs were assigned to flagellar assembly (654 upregulated and 75 downregulated genes) (Supplementary Table 3). Among the upregulated bacterial chemotaxis DEGs, we observed a strong enrichment of

genes expressing methyl-accepting chemotaxis proteins (MCP; fold change = 6.14) and purine-binding chemotaxis protein CheW (fold change = 4.70), which showed the highest expression in the dataset (Fig. 5c). The predominant flagellar assembly DEGs were genes encoding the flagellin FliC that functions in flagellar-mediated chemotactic motility, which showed a 3.66-fold enrichment in the salt-treated rhizosphere soil compared with the control (Fig. 5d). These results suggest that salt stress changes the rhizosphere community to be dominated by bacteria expressing for cell motility genes.

To determine the microbial populations that contributed to the increased expression of bacterial chemotaxis and flagellar assembly, we performed an assignment analysis of associated DEGs. This analysis revealed that the salt-induced increase in cell motility was mainly attributed to *Pseudomonas* (Fig. 5e, f). By sequencing the genomes of two *Pseudomonas* isolates XN05-1 and YE17 (Supplementary Fig. 11), we found that they harbored a series of motility related genes, including flagellar (*fli*, *flg* and *flh*) and motility (*mot*) genes, required for flagellar assembly (Supplementary Data 2). Additionally, strains XN05-1 and YE17 genomes both contained two main gene clusters of bacterial chemotaxis systems, and they included three genes encoding CheW, i.e., *cheW1*, *cheW2*, and *cheW3* (Supplementary Data 2 and Supplementary Fig. 12). Overall, above data indicate that salt stress alters the transcriptional activity of the rhizosphere microbiome, especially cell motility, and the change in rhizosphere function might be driven by *Pseudomonas* species.

### Root exudate component xanthine reproduced *Pseudomonas* enrichment

Root exudates have been reported to mediate the beneficial bacteria recruitment under biotic and abiotic stresses[24]. To identify potential key metabolites that induce *Pseudomonas* enrichment in salt-stressed wild soybean, root exudates were collected and conducted untargeted metabolome sequencing. In total, 658 metabolites were identified, the majority of which were lipids, flavonoids, organic acids, and phenolic acids (Supplementary Fig. 13). Many metabolites ($n = 317$) were significantly enriched in the salt treatment groups. Metabolites correlated with *Pseudomonas* genus from the amplicon data were investigated, and 175 metabolites showing positive (173) or negative (2) correlations (absolute value of Pearson's correlation > 0.9, $P < 0.01$) were identified (Fig. 6a). Among the metabolites showing a positive correlation with *Pseudomonas*, N-cyclohexylformamide, xanthine, and 2,4,5-trimethoxybenzoic acid were 6.64-, 6.07-, and 5.30 $\log_{10}$-folds more abundant respectively, in salt treatment group than in the control group and were the top three enriched metabolites (Supplementary Fig. 14 and Supplementary Table 4). These compounds were hardly measured in the control but were secreted by salt-treated plants in large quantities (Supplementary Fig. 15), suggesting that they were specific compounds correlated with salt stress.

Since the xanthine content was proportional to the salt stress strength (Supplementary Fig. 15) and it is harmless to humans and commercially available, we next focused on xanthine to investigate if it involved in *Pseudomonas* enrichment. We first inoculated a mixed culture of *Pseudomonas* strains YE17 and XN05-1 into sterile soil, and planted wild soybean seedlings. The result showed a higher absolute abundance of *Pseudomonas* in the xanthine supplementation groups than that in the control group (Supplementary Fig. 16). To further test if the *Pseudomonas* enrichment in the roots after xanthine application was a genus-specific trait, xanthine was added into natural soil growing wild soybean seedlings. The rhizosphere soil and root samples were collected, and 16S rRNA amplicon sequencing was performed to investigate the enriched microbes by xanthine. We found that xanthine clearly altered the bacterial communities in the root and rhizosphere soil (Fig. 6b). Intriguingly, among the top five most abundant genera, *Pseudomonas* was enriched in the xanthine addition group with highest fold change, which was up to 21.5-fold higher in the xanthine treated rhizosphere soil (relative abundance of 17.47%) than that in the control (0.81%) (Fig. 6c and Supplementary Fig. 17). The results indicate that xanthine-induced bacterial enrichment is largely specific to *Pseudomonas* genus. Above findings also indicate that exogenous xanthine addition reproduced the result of *Pseudomonas* enrichment within wild soybean root observed under salt stress condition, further supporting that root-secreted xanthine plays an important role in *Pseudomonas* recruitment.

To validate if the *Pseudomonas* enrichment correlated directly with the enhanced salt stress tolerance in wild soybean, we performed experiments involving *Pseudomonas* inoculation devoid of xanthine, applying xanthine but lacking *Pseudomonas* and inoculating *Pseudomonas* in the presence of xanthine. The results showed that xanthine addition alone slightly, but not significantly, enhanced plant growth, which was further enhanced in the presence of *Pseudomonas* (Fig. 6d). This result shows that wild soybean growth promotion after xanthine application is dependent on *Pseudomonas*.

### The role of motility related gene *cheW* in the chemotaxis toward purine and plant salt tolerance

As we mentioned above, the expression of motility related genes (especially the purine-binding chemotaxis protein CheW) was increased under salt stress; thus, we hypothesized that the motility related gene *cheW* plays an important role in *Pseudomonas* enrichment induced by purine, such as xanthine. To test this hypothesis, we first evaluated the effect of xanthine on the chemotaxis of *Pseudomonas* isolates. The results indicated that both strains XN05-1 and YE17 showed strong chemotaxis toward xanthine (Supplementary Fig. 18a, b). Moreover, quantitative chemotaxis assays of the two *Pseudomonas* isolates toward other purine or its derivatives, e.g., 6-benzylaminopurine, 2-aminopurine, hypoxanthine, and guanine, were also conducted, as these compounds were also secreted in higher amounts under salt stress (Supplementary Table 5). The results indicated that strains XN05-1 and YE17 both showed chemotaxis toward most of these compounds (Supplementary Fig. 18c, d).

To further verify the role of CheW in the *Pseudomonas* chemotaxis and plant salt tolerance enhancement, we deleted three *cheW* genes from strains XN05-1 and YE17 using the tri-parental conjugation method to obtain the Δ*cheW1*, Δ*cheW2*, and Δ*cheW3* strains of XN05-1 and YE17. We found that the strains XN05-1Δ*cheW3* and YE17Δ*cheW3* still showed obvious chemotaxis toward xanthine. However, XN05-1Δ*cheW1* and YE17Δ*cheW1* completely lost the chemotactic behavior to xanthine, whereas that in XN05-1Δ*cheW2* and YE17Δ*cheW2* was impaired (Fig. 6e, f). We next used the mutant strains to inoculate wild soybean seedlings under salt stress, and found that Δ*cheW3* mutant, the one still having chemotaxis ability, displayed the same extent of plant biomass as the wild-type strains. However, Δ*cheW1* and Δ*cheW2* mutants of XN05-1 and YE17 both cannot show plant biomass enhancement (Fig. 6g). This highlighted the critical role of motility related gene *cheW* in the interaction between the host and root-associated *Pseudomonas*. In summary, our results reveal that under salt stress condition, wild soybean exudate specific metabolites (e.g., xanthine) and affect the root microbiota to attract *Pseudomonas* species, which in turn promotes plant growth under salt stress (Fig. 7).

## Discussion
Here we characterize the salt induced root microbiome shifts in wild soybean. Under salt stress, we identified many enriched and depleted microbes, and demonstrated that salt led to relatively stronger microbial enrichment in the root compared with rhizosphere and bulk soil compartments, with highly preferential enrichment for *Pseudomonas*, which was evidenced to promote wild soybean growth in this study. Finally, the interaction between root exudates and altered root-associated microbiota under salt stress were elucidated.

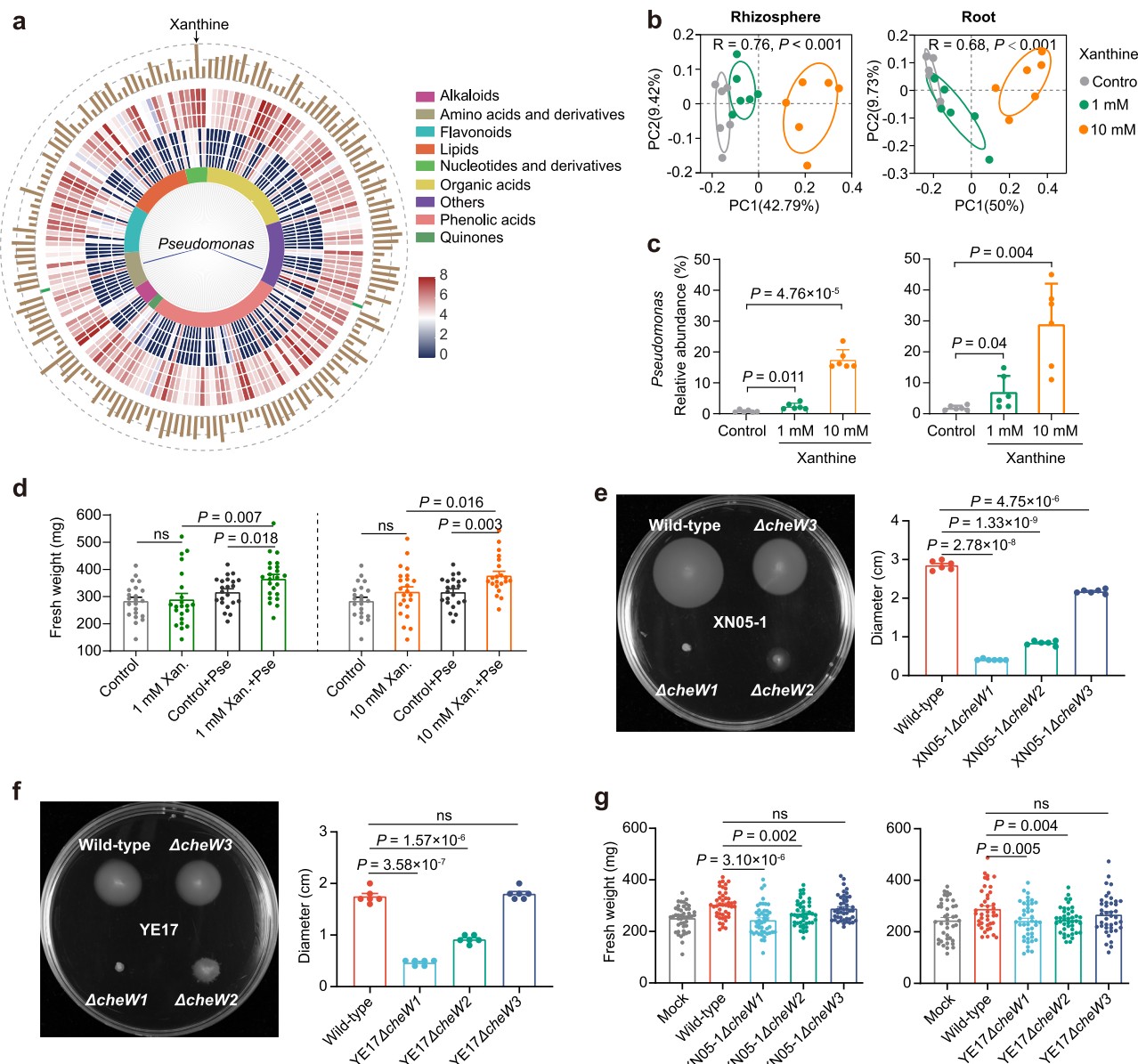

**Fig. 6 | The application of root exudate component xanthine enriched *Pseudomonas*.** **a** The Pearson's correlation of *Pseudomonas* genus (from 16S rRNA gene amplicon data) and root exudates. The inner lines indicate a statistically significant (Student asymptotic *p*-value < 0.01) correlation with coefficient < −0.9 (negative correlation: blue lines) or > 0.9 (positive correlation: other lines). The inner colored ring represents the classification of each compound. The middle heatmap is the abundance of compound in three control (inside) and salt-treated roots (200 mM NaCl; outside). The outer ring of bars indicates the log₂-fold enrichment (brown bars) or depletion (green bars) of each compound within salt-treated samples compared with control samples. **b** PCoA ordination of root and rhizosphere soil microbiota between control and xanthine applications (1 and 10 mM). Statistical analysis was performed using ANOSIM (analysis of similarities). **c** The relative

abundance of *Pseudomonas* genus in root and rhizosphere soil between control and xanthine applications (1 and 10 mM). *n* = 6 biologically independent samples. **d** Fresh weight between control, xanthine application alone, *Pseudomonas* inoculation alone and xanthine applications in presence of *Pseudomonas*. *n* = 22 plants per treatment except for 1 mM Xan.+Pse and 10 mM Xan. (*n* = 23 plants), and for 10 mM Xan.+Pse (*n* = 21 plants). **e**, **f** The chemotaxis of wild-type strains and *cheW* mutants of strains XN05-1 (**e**) and YE17 (**f**) response to 1 mM xanthine. *n* = 6 biologically independent samples. **g** Fresh weight of wild soybean inoculated with wild-type strains or *cheW* mutants of strains XN05-1 and YE17. *n* = 45 plants for strain XN05-1 and *n* = 41 plants for strain YE17. Values are means ± SEM for (**c**–**g**). Significance (**c**–**g**) was determined using two-sided Student's *t*-test *P* < 0.05. ns, not significant. Source data are provided as a Source Data file.

Our results are similar to previous studies declaring that the diversity and composition of root communities are more strongly influenced by abiotic stresses, relative to rhizosphere and bulk soils[16,34]. A closer interaction between the endophytic microbes and their hosts may contribute to this differential response. Numerous recent studies have revealed that biotic and abiotic stresses could recruit beneficial bacteria that enhance plant stress tolerance and promoting plant growth[4,12,14]. Our amplicon sequencing data revealed that the relative abundance of Pseudomonadales, especially the genus *Pseudomonas*,

increased with different salt stress levels (Figs. 2 and 3). Similarly, our metagenomic and metatranscriptomic data both showed an increase in the relative abundance of *Pseudomonas* under salt stress (Fig. 3e, f). *Pseudomonas* enrichment was also observed in domesticated soybeans (Supplementary Fig. 6), and salt-sensitive and salt-tolerant plants belonging to the family Curcurbitaceae under salt stress[15]. The presence of *Pseudomonas* across multiple plants indicated this genus might establish mutualistic interaction with their hosts[35]. It is important to evaluate if this close association is consistent across a broader

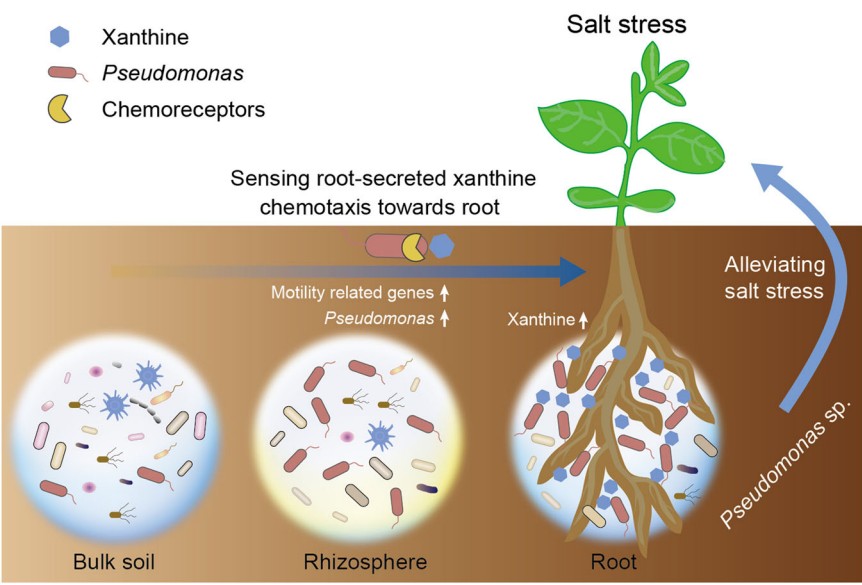

**Fig. 7 | Schematic drawing illustrating the recruitment of *Pseudomonas* by wild soybean under salt stress.** Soil *Pseudomonas* was attracted by xanthine secreted by stressed plant, and moved toward rhizosphere through chemotactic motility.

range of host plants and soil types. Nevertheless, we revealed for the first time that *Pseudomonas* was the dominant and most active member in the salt-treated rhizosphere soil.

Two *Pseudomonas* OTUs, OTU2336 and OTU4227 were prevalent in salt-stressed plants, with and without niche preference, respectively (Fig. 3g). Although we did not get isolates resembling OTU4227, it is important that future studies compare the functions of isolates with and without niche preferences for plant growth promotion. In this study, two corresponding *Pseudomonas* isolates YE17 (resembling OTU2336) and XN05-1 were selected, and they significantly increased growth of salt-stressed wild soybean (Fig. 4). A variety of *Pseudomonas* species, e.g., *P. stutzeri*, *P. koreensis* and *P. fluorescens*, are potentially beneficial to host plants under salt stress[36–39] due to their significant properties in improving compatible solutes, antioxidant status, and plant growth[40,41], all of which may result in their widespread occurrence in plants under stressful environments. It should be noted that strain XN05-1 shows the highest similarity with *Stutzerimonas frequens* (99.65%) and *Stutzerimonas stutzeri* (99.58%) now. *Stutzerimonas* is a recently proposed novel genus, and species in this genus formerly belongs to the *P. stutzeri* phylogenetic group[42,43].

To analyze salt-induced functional shifts in the rhizosphere microbiota, we conducted metagenomic and metatranscriptomic sequencing. According to the COG annotation, cell motility was enriched under salt stress in both datasets (Fig. 5a, b). Cell motility, especially bacterial chemotaxis, plays an important role in plant-microbiome signaling pathways[44]. In this study, genes encoding MCPs and a series of chemotaxis proteins (especially CheW) were upregulated in the salt treatment. Similar enrichment of chemotaxis genes has been observed under other types of stresses[45]. MCPs are the most common chemoreceptors in prokaryotes that sense chemical cues and regulate cellular activities including biofilm formation and flagellum biosynthesis[46]. CheW is a coupling protein at the core of the chemotaxis signaling network that connects chemoreceptors to CheA kinase[47]. Intriguingly, we found that the upregulated cell motility related genes were mainly derived from the salt-responsive taxon *Pseudomonas* (Fig. 5e, f).

Root exudates comprise a wide range of compounds, including organic acids, amino acids, sugars, nucleotides, and flavonoids[48,49], and mediate the interactions between plants and rhizobacteria[25]. Previous studies observed increased concentrations of flavonoids in the root

exudates of *G. max* (soybean) and *Phaseolus vulgaris* (common bean) grown under salt stress condition[50,51]. It had also been reported that amino acids (e.g., proline)[52,53], and organic acids (e.g., malic acid and citric acid)[54,55] were secreted in large quantities when plants were subjected to salt stress. However, in our study, these metabolites were either depleted or enriched with a very low fold change. Whereas, xanthine was increased in the root exudate of wild soybean with the second highest $\log_{10}FC$ value in this study. Xanthine is a purine derivative containing a purine base and is a product of purine degradation pathway. Other purine derivatives, including 6-benzylaminopurine, 2-aminopurine, hypoxanthine, and guanine, were also detected in this study and significantly enriched in the salt-treated roots (Supplementary Table 5). Purines are direct precursors for synthesizing cofactors and essential components of plant hormones that regulate plant growth and development[56,57]. Some purine derivatives, e.g., benzyl amino purine, were reported as plant growth enhancers[58]. Furthermore, purine metabolites are a continuous source of nitrogen for plant growth, especially in legumes[59,60].

Consistent with previous findings that purine derivatives could mediate *Pseudomonas* chemotaxis[61,62], our results also observed that representatives of the salt-responsive taxon *Pseudomonas* XN05-1 and YE17 showed strong chemotaxis toward purine and its derivatives. The addition of exogenous xanthine to natural soil markedly stimulated *Pseudomonas* enrichment in wild soybean root. Moreover, we found that *cheW* mutants of *Pseudomonas* XN05-1 and YE17 lost the chemotactic behavior toward root exudate xanthine. This demonstrated that purine, especially xanthine, mediated the salt-induced "cry for help" to enrich plant growth-promoting bacteria *Pseudomonas* by stimulating its motility. Similar phenomenon has been observed by previous works. For example, the flavanone naringenin induced chemotaxis in *Aeromonas* sp. H1 with enhanced bacterial motility and improved plant dehydration tolerance[63]. However, we cannot eliminate the possibility that other root exudates also contribute to the microbial community shift under salt stress. Further studies are required to unravel if and how purine interact with other compounds to regulate *Pseudomonas* enrichment.

Taken together, we present the first study on the salt-affected root microbiome of wild soybean using metagenomics, metatranscriptomics, and metabolomics approaches, along with downstream controlled experiments to validate the hypotheses generated from the

multiomics data. Our results show that salt induces a much stronger shift in the root-associated microbiota than in the bulk soil microbiota. *Pseudomonas* is the dominant stress-responsive taxon, and its corresponding isolates could promote wild soybean growth under salt stress, which supports the previously reported "cry for help" theory[4]. Root exudate purine or its derivatives, especially xanthine, increase the motility and colonization of *Pseudomonas*. This is consistent with the higher expression of genes involved in cell motility observed in the salt treatment than in the control. Thus, we conclude that the root exudate component purine attracts soil *Pseudomonas* into the root and benefit plant growth under salt-stressed condition.

## Methods

### Experiment design
To characterize the effect of salt on the root-associated communities and functions, we grew wild soybean plants in agricultural soil (pH 8.42, EC 337 µs/cm, TC (%) 4.63, TN (%) 0.76, organic matter 21.6 mg/kg) under controlled conditions. Briefly, wild soybean seeds were surface sterilized with 0.15% mercuric chloride for 10 min and washed 6 times with sterile water. Thereafter, the seeds were germinated in Petri dishes with sterile water in a growth chamber. Seedlings were transferred to plastic pots containing 200 g of field soil, and incubated at 25 °C on a 16 h/8 h light/dark cycle. After 10 days of wild soybean growing in pots, we treated them with three salt concentrations (i.e., 100, 200 and 300 mM NaCl), and sterile water without NaCl was used as control group. The experiment was performed using four biological replicates (six wild soybean plants in each pot generated one replicate) for all treatments. To determine the dynamic changes of root-associated microbiome affected by salt stress, we collected samples at three-time points: 1 day, 7 days and 14 days after salt stress.

### Sampling
Rhizosphere soils of six plants in each pot were combined to generate one composite sample, as were the root samples. Root-adhered soil was washed, the washing buffer was then centrifuged, and the resulting pellet was defined as rhizosphere soil. To collect root samples, roots were further washed and sonicated. After removing washing buffer, they were frozen[64]. We referred to the unplanted soil as bulk soil, which were collected from ~3 cm below surface. All samples were stored at −80 °C until DNA extraction. Samples including four treatments (0, 100, 200 and 300 mM NaCl), with three compartments (bulk soil, rhizosphere soil and root) and three time points (1 day, 7 days and 14 days) and four replicates for each were collected. The summary of sampling information was shown in Supplementary Data 3.

### Measurement of the soil properties
The soil EC was determined in a mixture with soil: water ratio of 1:5 (w/v) using EC meter (DDSJ-308A, Leici, China). The soil total carbon (TC) and total nitrogen (TN) concentrations were measured using a CHNS/O elemental analyzer (FlashSmart, Thermo Fisher, Germany).

### DNA extraction, 16S rRNA gene amplicon sequencing and data processing
Genomic DNA extraction for all samples was performed using the FastDNA® Spin Kit for Soil (MP Biomedicals) according to the manufacturer's protocol. The concentration of extracted DNA was measured with the NanoDrop spectrophotometer (ND2000, Thermo Scientific, DE, USA). Amplification of the V5-V7 regions of the 16S rRNA gene was performed using the primers 799F (5'- AACMGGATTAGATACCCKG-3') and 1193R (5'- ACGTCATCCCCACCTTCC-3'). The PCR reaction was performed in triplicate at 95 °C for 3 min, followed by 27 cycles of 95 °C for 30 s, 55 °C for 30 s, 72 °C for 45 s and a final extension step of 72 °C for 10 min. PCR products were purified and mixed in equal density ratios. Sequencing libraries were generated using the NEXTFLEX Rapid DNA-Seq Kit (Bioo Scientific, USA) following the manufacturer's

recommendations and sequenced on the Illumina MiSeq PE300 platform at the Majorbio Bio-Pharm Technology. Analysis of the 16S rRNA gene sequences was performed as described in our previous study[5]. Briefly, quality-filtered sequences were clustered into OTUs with 97% sequence similarity using UPARSE[65]. The representative sequence for each OTU was taxonomically classified with the RDP Classifier 2.2[66] and annotated against the SILVA database (release 138). All OTUs identified as chloroplast and mitochondria were discarded from the data set.

### Metagenomic sequencing
Rhizosphere samples collected at 14 days after salt stress were performed for metagenomic shotgun sequencing. Libraries were prepared without any amplification step, and sequencing was conducted on the Illumina HiSeq X-Ten platform, with 2 × 150 bp paired-end reads at the Majorbio Bio-Pharm Technology. The reads that contain adapters, low quality bases and 10% of undefined bases were removed. A total of ~264 Gb clean data was obtained from 16 samples. Megahit software was used to assemble high quality reads, and the obtained contigs shorter than 300 bp were discarded. After gene prediction with MetaGene, genes were clustered to remove redundant sequences using CD-Hit[67] at the 90% identity and 90% coverage. To perform taxonomic and functional analysis, the genes were compared (BLASTp) against NCBI-nr, KEGG and COG databases using DIAMOND[68] with an e-value cutoff of $10^{-5}$.

### RNA extraction and metatranscriptomic sequencing
For metatranscriptomic sequencing, independent rhizosphere samples ($n = 3$ biological replicate) were prepared according to above description. One biological replicate (> 5 g) was generated from 60 control plants or 120 salt-treated plants as only small amount of sample was collected from stressed plants. Total RNA was extracted using TRIzol® Reagent according the manufacturer's instructions (Invitrogen, USA). DNase I (TaKara, Beijing, China) was used to remove genomic DNA. Metatranscriptome libraries were constructed according to the TruSeqTM Stranded Total RNA Sample Preparation Kit (San Diego, CA) with 5 µg of high-quality RNA. All samples were sequenced in the Illumina HiSeq 2500 instrument at Shanghai Biozeron Biothcnology Co., Ltd. (Shanghai, China). The raw reads were trimmed and quality controlled by Trimmomatic v0.36[69]. Then the clean reads were aligned to the genome of wild soybean, the SILVA SSU (16S/18S) and SILVA LSU (23S/28S) databases to remove host genome and rRNA related reads, respectively. A total of ~71 Gb clean data was obtained from six samples. Clean data were assembled with megahit software. After gene prediction by METAProdigal, genes were clustered to remove redundant sequences using CD-Hit[67] at the 95% identity and 90% coverage. All genes searched against the NCBI-nr, COG and KEGG databases using BLASTp with an e-value cutoff of $10^{-5}$. Salmon (https://github.com/COMBINE-lab/salmon) was used to calculate the expression level for each transcript. EdgeR (Empirical analysis of Digital Gene Expression in R) was applied to identify DEGs between control and salt treatment, with the cutoff of fold change > = 2, and false discovery rate (FDR) adjusted $P$ < = 0.05.

### Root exudates collection and metabolomic sequencing
Wild soybean seeds were surface sterilized and germinated in Petri dishes at 25 °C and 70% relative humidity. Two days old seedlings were planted in soil and incubated at 25 °C with a 16 h/8 h light/dark cycle. After 14 days, these seedlings were washed with sterile water to remove all root attached soil. These seedlings were transferred to the hydroponic system (containing 0, 100, 200 and 300 mM NaCl solution) for root exudates collection. We found all plants in 300 mM solution died due to excessive NaCl concentration, therefore, two salt treatments (100 and 200 mM) were remained. Each treatment contained three replicates and each replicate consisted of 20 seedlings. The water containing exudates was filtered with 0.22 µm filters to remove root

debris and microorganisms. Three replicates were conducted for each treatment. Since the root exudates were collected from seedlings growing in sterile hydroponic system, the possibility that compounds derived from microorganisms could be excluded. The collected root exudates were analyzed using an UPLC-ESI-MS/MS system and Tandem mass spectrometry system at Wuhan MetWare Biotechnology Co., Ltd. (China). Metabolites were identified using both mass spectrum and retention time. To compare the content of each metabolite between the control and treatments, the mass spectrum peak of each metabolite in different samples was calibrated according to integration of the peak area.

## qPCR for total bacteria and *Pseudomonas*

Total bacterial abundance was determined using the Eub338F/Eub518R primers[70] (Supplementary Table 6). Standard curves were generated using 10-fold serial dilutions of a plasmid containing the 16S rRNA gene from *Arthrobacter pokkalii*[71]. The specific primer Ps-for and Ps-rev[72] (Supplementary Table 6) was used to quantify *Pseudomonas* densities. Standard curves were generated using 10-fold serial dilutions of a plasmid containing the 16S rRNA gene from *P. stutzeri*. The qPCR analyses were performed using an Applied Biosystems 7500 Real Time PCR System (Applied Biosystems, USA). Each assay was performed in triplicate.

## Bacterial isolation and greenhouse experiment

Root samples from salt-treated plants were ground and spread on TSA (tryptic soy agar) and R2A (Reasoner's 2A agar) media for bacterial isolation. All plates were incubated at 28 °C for 2–4 days. Bacterial colonies were picked, purified for three times, and stored at −80 °C with 15% (v/v) glycerol. The bacterial genomic DNA was extracted using an EasyPure Bacteria Genomic DNA Kit (TransGen, Beijing, China) according to the manufacturer's instructions. Bacterial universal primers 27F/1492R[73] (Supplementary Table 6) were used to amplify the full-length 16S rRNA genes. The sequences were submitted to EzBioCloud (www.ezbiocloud.net/)[74] for bacterial identification.

For the greenhouse experiment, *Pseudomonas* isolates XN05-1 and YE17 were conducted independently. Briefly, wild soybean seeds were surface sterilized with 0.15% mercuric chloride and germinated in Petri dishes with sterile water. Three days old seedlings were transplanted to a plastic pot. For each treatment, ten pots were prepared with two seedlings per pot. *Pseudomonas* isolates XN05-1 and YE17 were cultivated using nutrient broth medium for overnight and centrifuged at 4 °C, 5000 × g to concentrate bacterial cells, and then resuspended them in the sterile water. After five days of transplantation, bacterial cells were inoculated using root drenching methods with a final density of ~10⁷ cells per gram of soil. *Pseudomonas* was inoculated for three times at intervals of three days. The control group was treated with equal amounts of sterile water. Wild soybean plants were incubated in a climate chamber at 25 °C and watered regularly with sterile water. Finally, growth phenotypes, including shoot and root length and fresh weight, were measured.

## *Pseudomonas* colonization

To investigate the *Pseudomonas* colonization with and without salt stress, wild soybean seeds were surface sterilized with 0.15% mercuric chloride and germinated in Petri dishes with sterile water. Three days old wild soybean seedlings were transplanted to a plastic pot containing 200 g of sterile soil. The mixed culture of two *Pseudomonas* isolates XN05-1 and YE17 (OD$_{600}$ = 0.3) were inoculated in planted soil after seven days. Then, 200 mM NaCl was applied, and sterile water was used as control. These plants grew in a growth chamber under 25 °C on a 16 h/8 h light/dark cycle for two weeks. The *Pseudomonas* abundance within root samples were quantified by qPCR using specific primers (detail methods see the section of qPCR for total bacteria and *Pseudomonas*).

## Genome sequencing

Strains XN05-1 and YE17 were grown in NB (nutrient broth) medium for overnight and genomic DNA was extracted with the SDS method[75]. The whole genome was sequenced using PacBio Sequel platform and Illumina NovaSeq PE150 at the Beijing Novogene Bioinformatics Technology Co., Ltd. SMRT Link v5.0.1 software[76] was used for genome assembly. A whole genome Blast search (E-value < 1e-5, minimal alignment length percentage > 40%) was performed against KEGG database.

## *cheW* mutants construction of *Pseudomonas*

The *cheW* gene deletion mutants were constructed using tri-parental conjugation with the suicide vector pK18mob*sacB*[77]. The primers used are shown in Supplementary Table 6. Detailly, upstream and downstream fragments of strains XN05-1 and YE17 genomic DNA was amplified using the primer pairs XN05-1-cheW1-UF/UR and XN05-1-cheW1-DF/DR, respectively. Both fragments were purified and inserted into the suicide vector pK18mobsacB. The obtained plasmid was transformed into *E. coli* DH5α, and verified by PCR and subsequent Sanger sequencing. Then the correct plasmid pK18mob*sacB* was transferred into XN05-1 Amp$^R$-derivative strain by tri-parental conjugation using the helper plasmid pRK2013[78]. The mutant candidates resistant to gentamicin (25 μg/ml) and ampicillin (50 μg/ml) were used for a PCR screen and sequencing. The potential mutant was performed sucrose counterselection using NA medium containing 15% (w/v) sucrose to cause the excision of the suicide vector from the chromosome. The resultant mutant was confirmed by PCR with the primer pair XN05-1-cheW1-UF/XN05-1-cheW1-DR and sequencing, named XN05-1Δ*cheW1*. The other mutants (XN05-1Δ*cheW2* and XN05-1Δ*cheW3*) of XN05-1 and three *cheW* mutants (YE17Δ*cheW1*, YE17Δ*cheW2* and YE17Δ*cheW3*) of YE17 were constructed in the same way using their respective primers.

## Quantitative chemotaxis assays

Assays were carried out according to previous methods[79] with minor modifications. Briefly, *Pseudomonas* isolates were cultivated in M9 minimal medium supplemented with 1% glucose, and cultures were grown to an OD$_{600}$ of ~0.4. Cells were washed twice by centrifugation and resuspended in chemotaxis buffer (30 mM K$_2$HPO$_4$, 19 mM KH$_2$PO$_4$, 20 μM EDTA and 0.05% (v/v) glycerol, pH 7.0) to an OD$_{600}$ of 0.1. Aliquots (200 μl) of the bacterial suspension were transferred into the wells of 96-well microtiter plates. Then, 1 ml syringes were filled with 100 μl of chemotaxis buffer (negative control) or chemoeffector solution (1 mM). Then, syringe needles were immersed into the cell suspension. After incubation for 30 min, the syringe was removed from the cell suspension, rinsed with sterile water, and emptied into an Eppendorf tube. Cultures were serially diluted with 10-fold, and then 5 μl dotted on agar plates. Colonies were counted after growth for 48 h at 28 °C. Data shown are from three biological replicates.

Quantitative soft agar plate assay was used to access chemotaxis of wild-type strains and *cheW* mutants of *Pseudomonas*. This method is based on a chemical gradient created by the bacterial consumption for the attractant, which can activate the chemotactic response[77]. We selected this assay here as it could compare the chemotactic behaviors of wild-type and mutant strains visually. All testing strains were grown overnight in NB medium. The cultures were harvested by centrifugation, then washed and resuspended in a minimal medium to an OD$_{600}$ of 1.0. Aliquots (2 μl) of the suspensions were inoculated onto minimal soft agar plates containing 1 mM xanthine and 0.3% agar. Plates were incubated at 28 °C for 24 – 48 h, then the diameters of all mutant and wild-type strains were measured. Data shown are from six biological replicates.

## Exogenous xanthine application

The culture of two *Pseudomonas* isolates XN05-1 and YE17 were mixed with sterile soil to a final density of ~10⁷ cells per gram of soil. Soil

without *Pseudomonas* inoculation was used as control to study the effect of xanthine on plant growth. Three days old wild soybean seedlings were transplanted to a plastic pot containing 50 g of above soil. After ten days, 5 ml of two concentrations (1 and 10 mM) of xanthine solution were added in root surrounding soil, and the solution supplied once every day for three days. These plants grew in a growth chamber under 25 °C on a 16 h/8 h light/dark cycle for two weeks. Then fresh weight of plants was measured.

Natural unsterile soil was used for study the bacterial community change after exogenous xanthine application (1 and 10 mM). Rhizosphere soil and root samples were collected for 16S rRNA gene amplicon sequencing.

## Statistical analyses

ANOSIM was used to evaluate the differences among the microbial communities of different treatments and rhizocompartments. PERMANOVA was calculated based on Bray-Curtis distance with 999 permutations using the vegan package[80] of R version 3.5.3. PCoA was performed to ordinate the microbial composition in the different samples based on Bray-Curtis distance with the vegan and ggplot2 packages in the R software. Genus's abundance from control sample was used as a control to calculate the enrichment ($P < 0.05$; fold change > 2) or depletion ($P < 0.05$; fold change < 0.5) profile of salt treatments, which was illustrated with heatmap using TBtools[81]. DEGs were identified using edgeR with the logarithmic of fold change > 2 and FDR adjusted $P < = 0.05$. Pearson correlation analysis between microbial community and metabolites was calculated using the cor function of R. All statistical analyses were performed using two-sided $t$-tests, unless otherwise specified.

## Reporting summary

Further information on research design is available in the Nature Portfolio Reporting Summary linked to this article.

## Data availability

Raw 16S rRNA gene amplicon sequence and metagenomic data derived from salt stress experiments are publicly available under NCBI BioProject number PRJNA890906. Metatranscriptomic data has been deposited in the Sequence Read Archive under BioProject number PRJNA989423. All 16S rRNA sequence data derived from xanthine application experiments are publicly available under NCBI BioProject number PRJNA1064605. The genomic data for strains XN05-1 and YE17 are publicly available under NCBI BioProject number PRJNA1061746. The SILVA database is available at https://www.arb-silva.de/. The NCBI-nr database is available at https://www.ncbi.nlm.nih.gov/. The KEGG database is available at http://www.genome.jp/kegg/. The eggNOG database is available at http://eggnog6.embl.de/#/app/home. Source data are provided with this paper.

## Code availability

The scripts used in this study are available at GitHub (https://github.com/helloacc/wildsoybean_microbiome).

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

## Acknowledgements

Y.Z. was supported by grants from the National Natural Science Foundation of China (32101289) and the Shandong Provincial Natural Science Foundation (ZR2021QD026). C.-S.Z. and Z.X. were supported by a grant from the National Natural Science Foundation of China (U1806206), C.M. was supported by a grant from the National Natural Science Foundation of China (32171948). C.-S.Z. was additionally supported by a grant from the Agricultural Science and Technology Innovation Program of China (ASTIP-TRIC-ZD04).

## Author contributions

C.-S.Z. designed the experiments, analyzed the data, and wrote the manuscript. Y.Z. designed the experiments, analyzed the sequence data, wrote the manuscript, and prepared most figures and tables. X.C. performed the bioinformatic analyses and visualized the data. Y.Zhou conducted wild soybean salt stress, growth promoting experiment and qPCR. S.M. analyzed metabolomic data and discussed the results. Y.W. performed pot experiments and visualized the data. Z.L. and H.Z. conducted *cheW* mutants and pot experiments. Y.Y. performed chemotaxis assays and pot experiments. D.Z., C.M. and Z.X. discussed the results and provided interpretation of results. X.S. isolated *Pseudomonas* strains. K.X. and Y.L. provided critical suggestions. All authors edited and approved the manuscript.

## Competing interests

The authors declare no competing interests.
