## [Peer Review File · Nature Communications]

Reviewers' Comments:

Reviewer #1:

Remarks to the Author:

Salt stress represents a formidable abiotic challenge with profound global implications, and its resolution hinges on attaining a comprehensive grasp of the intricate interplay between plants and their microbiomes. The topic undeniably holds paramount importance and possesses the potential to captivate a broad audience. Nonetheless, a series of notable concerns have surfaced that merit deliberate consideration along with potential remedies:

Major Concerns:

Pseudomonas Isolates Experiment: The derived conclusion from the *Pseudomonas* isolates experiment, which accentuates the role of inducing motility for salt stress response, seemingly contradicts the absence of motility-related genes in *Pseudomonas* strains Line251-256. Addressing this disparity necessitates a meticulous reexamination of the genome dataset, specifically focusing on the presence of motility-related genes within these isolates. Please add the motility genes discovery to the manuscript. While the present findings imply that xanthine triggers motility and its external application enriches *Pseudomonas* in non-stressed plants, further exploration is imperative to determine whether this enrichment correlates directly with heightened salt stress tolerance in soybeans. To validate this, pivotal experiments involving *Pseudomonas* inclusion devoid of xanthine and the inclusion of a control set featuring varying xanthine concentrations but lacking *Pseudomonas* are essential. As xanthine itself fosters growth, untangling its independent effects from those attributed to *Pseudomonas* is of utmost importance. Moreover, elucidating the precise role of motility in conferring salt stress resistance is pivotal. If feasible, introducing a motility knockout mutant could yield invaluable insights. Alternatively, advancing a discussion or hypothesis about the underlying advantages of motility in *Pseudomonas* survival and salt stress resistance, and subsequently incorporating this rationale into Figure 6, would serve to enhance clarity.

Selection Strategy for *Pseudomonas* Isolates: The strategy employed for selecting *Pseudomonas* isolates, as delineated in lines 186-188, assumes their independence from specific niches due to their prevalence in both root and rhizosphere samples. However, the selection of YE17, which closely resembles OTU2336 with distinct niche preferences, challenges this assumption. Addressing this would entail incorporating an additional isolate resembling OTU4227, which lacks niche preferences, for the inoculation experiments. Delving into the functional discrepancies in salt stress adaptation between isolates with and without niche preferences, and comprehending the reasons behind these variations, would greatly enrich the analysis.

Shotgun Metagenomics and NaCl Treatment: The utilization of shotgun metagenomics to assess 300mM NaCl-treated samples, while underscoring the importance of motility, might not be directly comparable to exudates from 100 and 200mM NaCl-treated samples due to disparate conditions. For coherence, it would be judicious to consider measurements of root exudates from 300mM NaCl-treated samples. Additionally, it is essential for the authors to explicitly state whether the current isolates possess genes pertinent to motility function.

Functional Shifts of *Pseudomonas* in MAGs: The presence of four condition-specific MAGs, illustrated in Figure 4g, and derived from co-assembled reads, MAG binning, and de-replication, has raised inquiries regarding the functional shifts of *Pseudomonas*. Given their concurrent enrichment under salt stress (supplementary figure 13), the argument for distinct functional shifts becomes less compelling. If there are *Pseudomonas* that didn't display enrichment under salt stress, a comparison between the four enriched MAGs and those that remained unenriched would be pertinent. If not, it might be advisable to omit this paragraph.

Minor Concerns:

Line23: I'd suggest to remove word "metabolites" from the manuscript. We could explore the enrichment pattern only with the microbiome data. Throughout the manuscript, careful revisions should be undertaken to ensure clarity and coherence.

Precision in Pseudomonas Enrichment Assertion: The assertion about salt-induced Pseudomonas enrichment in both root and rhizosphere (lines 218-219) necessitates a nuanced approach, considering that not all strains exhibit consistent enrichment patterns across various niches. Ensuring an accurate portrayal of this information is crucial.

Enhancement of Image Resolution Quality: Rectifying the issue of low image resolution quality across all figures is vital to guarantee clear and comprehensible visuals.

Appropriate Use of "Significantly": The term "significantly" (line 106) should be used precisely to reflect the nature of comparisons, taking into account that not all comparisons yield statistical significance.

Improved Figure Legends: Enhancing figure legends substantially to provide comprehensive descriptions would bolster the reader's comprehension of the visual content. For instance: It's unclear what "EC" represents in figure 1a. Are the labels "0," "100," "200," and "300" indicative of 0mM, 100mM, 200mM, and 300mM in figure 1d? Please provide clarification.

Readability of Supplementary Figure 2: Acknowledging the challenges posed by the color scheme in Supplementary Figure 2 and making necessary adjustments to enhance clarity and accessibility is recommended. For instance, addressing the similarity in color between Actinobacteria and Gemmatimonadita would improve differentiation.

Clarification of Diversity Drop in Figure 2a: The noticeable decrease in diversity for the 300mM NaCl-treated sample on Day 14 (Figure 2a) in contrast to other salt-treated samples warrants clarification. Providing context and explaining any distinct trends would contribute to a better understanding.

Elaboration on Abbreviations: Expanding on abbreviations such as "K," "N," etc. (line 285) would amplify reader comprehension.

By diligently addressing these major and minor concerns, the manuscript's lucidity, precision, and overall scientific rigor will undergo substantial improvement, leading to a more robust and impactful contribution to the domain of salt stress and plant microbiome interactions.

Reviewer #2:

Remarks to the Author:

This review concerns the manuscript "Purines enrich rhizosphere Pseudomonas to improve wild soybean growth under salt stress" by Yanfen Zheng et al.. The authors have studied wild soybean under salt stress and reveal an enrichment of Pseudomonas bacteria using cultivation independent methodology. They isolated 30 Pseudomonas strains and characterize two of them to enhance plant growth under salt stress. Rhizosphere metagenomics and metatranscriptomics reveal an enhancement bacterial cell motility features in a salty rhizosphere (including a purine-binding chemotaxis protein CheW). A metabolomic analysis reveals then an accumulation of purines in the rhizosphere. The two Pseudomonas strains then indeed respond chemotactically towards purines in invitro studies and application of purines enhances plant growth in presence of bacteria. Overall the authors present many interesting and possibly connected findings. However, the major problem is that the causal linkings of these findings were not demonstrated, while the authors report the findings as causally linked and also placing them into too generalized context. For demonstration of causal linkage, often genetic experiments (plant mutant not exuding purines and/or Pseudomonas mutants in CheW) proof helpful. Besides this major comment, there are a few further main concerns and numerous minor points that require attention.

Major comments:

Study context: The title claims that purines enrich Pseudomonas bacteria. While this is true for the 2 studied strains, it is not clear if this is a generic Pseudomonas- or strain-specific trait. Do other

Pseudomonas bacteria also enrich on roots after application of xanthine or is this a specific behavior of these two strains? Such a claim requires testing more generally *Pseudomonas* strains and also other bacterial genera.

Second, the title also claims that this enrichment leads to growth promotion. Again this has only been shown for the 2 strains without benchmarking other strains for generalization. Furthermore the xanthine-application experiments with the two strains lack an important control: how do the plants grow alone (without *Pseudomonads*!) after application of xanthine. This is necessary to answer if *Pseudomonas* bacteria mediate the growth promotion.

Also the third claim in the title - growth promotion under salt stress - is lacking an important control: The strain inoculation experiments were only performed under salt stress conditions. It remains unclear if the growth promotion is generic or specific to salt stress. Does it occur also under non-salty conditions.

Finally, the measurements of rhizosphere chemistry reveal an accumulation of purines in this compartment. Of course it is tempting to assume that they are exuded from plants. Can the authors exclude the possibility that the purine compounds have microbial origin?

The work is also placed in the context of the salt tolerant *G. soja* vs. ?salt-sensitive? *G. max*. The study covers data from both species, but rarely as a direct comparison. E.g. in Fig. 1 it would be nice to document the phenotype of *G. max*. Alternatively, the context of differential salt tolerance by these plants is not necessary.

Data connectivity: The connectivity of the different data often appears a bit occasional and should be improved. It would be important to better link the OTU data (representative OTU sequence) with the strains (full length 16S sequence (SANGER?), their genome) and the MAGs of the metagenomic sequencing. These different data share the same taxonomy assignment (Genus=*Pseudomonas*), but it is missed opportunity to compare the actual DNA sequences detected by the different approaches. This can be done simply by alignments and quantifying % sequence similarity. It is unclear if the isolated strains belong to the abundant MAGs. The credibility (how well the 2 strains can represent the culture independent findings) of data connection would greatly increase if these two isolates would fully correspond to the sequencings. A few further things appear a bit occasional such as why presenting the top 10 *Pseudomonas* (Fig. S7) compared to top 12 genera (Fig. 3).

RNA seq analysis: the fact that 77/189k genes vary in response to salt suggests a massive transcriptional reprogramming and then the following sentence mentions only fold changes of 1.4x and 1.6x (not reflecting a massive transcript response). I can hardly believe that so many genes vary with such a low fold change. I wonder about the number of replicates and the statistic threshold for this claim. FDR correction of p-values?

Metabolomics: Several aspects of the metabolomic analyses require clarification. To which type of *Pseudomonas* data (amplicon-genus, amplicon-OTU, Metagenome...) were they correlated to? The massive fold changes (log10!) suggest that these compounds were hardly measured in the control samples. Pls clarify coverage of control vs. treated samples and their normalization. Similar to OTU, metagenome or RNAseq data, also the metabolite data requires dual information on abundance and fold change for good interpretation. The authors have chosen dot plots in Fig. 3 for such dual information of the OTU data. Would it be possible to apply this principle also to the metabolite data? E.g. using MA plots?

Language: The manuscript requires editing by a native speaker. There are many 'linguistic bumps'. Furthermore some terminology needs to be clarified: is microbiota synonymous for microbiome? Difference between resistance and tolerance to salt stress? Consistent use of hyphen: root-associated, salt-induced... Attention to vocabulary, too: do not use terms without "direction". Example in L147: Instead of describing "shifts", specify whether you refer to and increase or decrease.

Figure legends are often insufficient or incomplete: A figure together with its caption should be self-explanatory. Detail the experiment and method with one sentence, the measurements must be explained, abbreviations defined, number of replicates (n = X) is missing throughout the manuscript. A caption shall also state the statistic test and eventual correction of P-values for multiple hypothesis testing. Figure 1 comes with a particularly rudimentary caption, no panel

indexes in the title.

Figure 3: The relative abundance plot in 3a-c is misleading. As it is it displays cumulative relative abundance of the 3 treatments. The 3 treatments should be shown side-by-side. Why not a dot plot with 4 columns (control, 100-300) where the dot size corresponds to relative abundance? To make the claim that root changes are strongest, it would be helpful to use the same scale for root, rhizosphere and soil. Since the work with the 2 *Pseudomonas* isolates is described in a separate paragraph, consider a separate main figure. Use boxplots instead of these non-scientific violine plots (individual replicates are poorly resolved).

Minor comments

L43: *many* strategies, and then only 2 examples follow

L47: what are "halophyte root associated environments"?

L50: Bacterial provision of nutrients should alleviate salt stress. It is hard to believe that this is the most important physiological process.

L75-78: rephrase

L99: EC based on available ions, no?

L101: precise that TC and TN are soil measures

Inconsistent usage salt treatments in text and figures. Sometimes used as

control/S100/S200/S300 or 0/100/200/300

L122/Fig2a: root 300 is not decreased at 14d

L125: refer to NMDS stress values.

L125/L128: demonstrate and indicate seem contradicting verbs for describing the same finding

L134...: The claims of that paragraph are not sustained with statistics in Figure S1. Is it useful at

all to mention a decrease of Proteobacteria, when the next sentence then reports increase of

Pseudomonadales and *Xanthomonadales*? High rank descriptions (Phylum or Order level) are often not very informative. I find the genus description sufficient.

L146: Why numerically mentioning the decrease of *Rhizobiales*, while not for the others? The story is about *Pseudomonadales*.

L151: it is unclear what these percent values (e.g. 21.74% of root genera) refer to. I cannot find this in Fig. S2 or Table S2. Cumulative relative abundance, fraction of genera?

L161: avoid such suggestive interpretation

L163: Enrichment of *Proteobacteria*? Inconsistent with L141

Fig. 2: endosphere or root?

L189: cite first Fig. 3a then 3b then 3c

L206: to discussion

L202: geographic distance is a poor discrimination for different soils. Use soil type definitions and/or chemical analyses instead.

L224: isolation from roots of salt stressed plants?

L237: show data of 'before' and 'after' salt stress

L243: equal culture?

L243: qPCR is a quantitative and not an observatory method

L249: The genome sequencing cannot be claimed to determine the genetic basis. It is simply descriptive for what genes these two strains have. There is not determination whether IAA, ACC... are involved here. How does the 16S sequence of the genome compare to the Sangersequence of the strain, the OTU and the MAG data?

L265/344: bule

L312: expressing, not encoding

Fig. 4g: I don't get the color link between MAGID and Treatment in the legend.

L322: which groups?

L354: why two and not three salt treatments?

Fig. 5a: would be better to display the fold changes as being an enrichment (+) or a depleting (-) and not both on same axis.

Figures 5d/e require important controls: how do the plants grow alone with application of xanthine (without *Pseudomonads*).

L447: I do not understand why these traits are discussed here. They do not link to the data of this study.

L479: The experiment had a time frame of 2 weeks – is this sufficient that HGT could explain

genome changes? Isn't it more likely that different strains become more abundant and this is reflected by the metagenomic analysis?

L557: are the four "biological" replicates pooled samples of six pots/plants. Are these 1 plant per pot or 6 (L563)?

Response to the editor and reviewers (NCOMMS-23-25309A)

Many thanks to the editor and reviewers for their constructive and helpful comments. We have thoroughly revised the manuscript in response to these comments (changes in the revised manuscript are in red). We greatly appreciate the time that went into providing such detailed reviews and hope we have addressed the concerns sufficiently. We now present our point-by-point responses to the reviewers:

Reviewer #1 (Response):

Salt stress represents a formidable abiotic challenge with profound global implications, and its resolution hinges on attaining a comprehensive grasp of the intricate interplay between plants and their microbiomes. The topic undeniably holds paramount importance and possesses the potential to captivate a broad audience. Nonetheless, a series of notable concerns have surfaced that merit deliberate consideration along with potential remedies:

We thank this reviewer for the interest in our results, and for these valuable suggestions. We present our point-by-point response to the comments below.

Major Concerns:

Pseudomonas Isolates Experiment: The derived conclusion from the *Pseudomonas* isolates experiment, which accentuates the role of inducing motility for salt stress response, seemingly contradicts the absence of motility-related genes in *Pseudomonas* strains Line251-256. Addressing this disparity necessitates a meticulous reexamination of the genome dataset, specifically focusing on the presence of motility-related genes within these isolates. Please add the motility genes discovery to the manuscript. While the present findings imply that purine triggers motility and its external application enriches *Pseudomonas* in non-stressed plants, further exploration is imperative to determine whether this enrichment correlates directly with heightened salt stress tolerance in soybeans. To validate this, pivotal experiments involving *Pseudomonas* inclusion devoid of xanthine and the inclusion of a control set featuring varying xanthine concentrations but lacking *Pseudomonas* are essential. As xanthine itself fosters growth, untangling its independent effects from those attributed to *Pseudomonas* is of utmost importance. Moreover, elucidating the precise role of motility in conferring salt stress resistance is pivotal. If feasible, introducing a motility knockout mutant could yield invaluable insights. Alternatively, advancing a discussion or hypothesis about the underlying advantages of motility in *Pseudomonas* survival and salt stress resistance, and subsequently incorporating this rationale into Figure 6, would serve to enhance clarity.

Thanks for your valuable suggestion.

--The original purpose of this part was to investigate the genes related to plant growth promotion in *Pseudomonas* strains. Thus, we did not show motility related genes. In

fact, the two *Pseudomonas* strains harbor a series of motility-related genes. We have listed them in Supplementary Data 2 and added description in Line 280-287.

Line 280-287: “By sequencing the genomes of two *Pseudomonas* isolates XN05-1 and YE17 (Supplementary Fig. 11), we found that they harbored a series of motility-related genes, including flagellar (*fli*, *flg* and *flh*) and motility (*mot*) genes, required for flagellar assembly (Supplementary Data 2). Additionally, *P. stutzeri* XN05-1 and *P. frederiksbergensis* YE17 genomes both contained two main gene clusters of bacterial chemotaxis systems, and they included three genes encoding CheW, i.e., *cheW1*, *cheW2*, and *cheW3* (Supplementary Data 2 and Supplementary Fig. 12).”

--We have performed new pot experiments including (1) inoculating *Pseudomonas* strains without xanthine; (2) providing different xanthine concentrations but lacking *Pseudomonas* strains; (3) inoculating *Pseudomonas* strains with different xanthine concentrations. The results showed that xanthine addition slightly, but not significantly, enhanced plant growth, and the promotion effect was further increased in the xanthine inclusion with the presence of *Pseudomonas*. This demonstrated that wild soybean growth promotion after xanthine application is dependent on *Pseudomonas*. We have added this results in our revised manuscript.

Line 331-338: “To validate if the *Pseudomonas* enrichment correlated directly with the enhanced salt stress tolerance in wild soybean. We performed experiments involving *Pseudomonas* inoculation devoid of xanthine, applying xanthine but lacking *Pseudomonas* and inoculating *Pseudomonas* in the presence of xanthine. The results showed that xanthine addition alone slightly, but not significantly, enhanced plant growth, which was further enhanced in the presence of *Pseudomonas* (Fig. 6d). This result shows that wild soybean growth promotion after xanthine application is dependent on *Pseudomonas*.”

--We have introduced *Pseudomonas* mutants in CheW. Three *cheW* genes were annotated in both two *Pseudomonas* genomes (Supplementary Fig. 12). Two *cheW* mutants (named $\Delta cheW1$ and $\Delta cheW2$) nearly lost or decreased the chemotactic ability to xanthine, emphasizing the importance of CheW in the interaction between *Pseudomonas* and root exudate purine. Furthermore, compared to wild type *Pseudomonas* isolates, the inoculation of *Pseudomonas* mutants ($\Delta cheW1$ and $\Delta cheW2$) lost the plant salt tolerance enhancement. Above results have been provided in Line 356-367 and Figure 6.

Line 356-367: “To further verify the role of CheW in the *Pseudomonas* chemotaxis and plant salt tolerance enhancement, we deleted three *cheW* genes from *P. stutzeri* XN05-1 and *P. frederiksbergensis* YE17 using the tri-parental conjugation method to obtain the $\Delta cheW1$, $\Delta cheW2$, and $\Delta cheW3$ strains of XN05-1 and YE17. We found that strains XN05-1 $\Delta cheW1$ and YE17 $\Delta cheW1$ lost the chemotactic behavior to xanthine, whereas that in XN05-1 $\Delta cheW2$ and YE17 $\Delta cheW2$ was impaired (Fig. 6e and f). We next used the mutant strains to inoculate wild soybean seedlings under salt stress, and found that $\Delta cheW3$ mutant showed the same extent of plant biomass as the wild-type strains. However, $\Delta cheW1$ and $\Delta cheW2$ mutants of XN05-1 and YE17 both led to a decreased

plant biomass compared with the wild-type strains (Fig. 6g). This highlighted the critical role of motility related gene *cheW* in the interaction between the host and root-associated *Pseudomonas*.”

Selection Strategy for Pseudomonas Isolates: The strategy employed for selecting *Pseudomonas* isolates, as delineated in lines 186-188, assumes their independence from specific niches due to their prevalence in both root and rhizosphere samples. However, the selection of YE17, which closely resembles OTU2336 with distinct niche preferences, challenges this assumption. Addressing this would entail incorporating an additional isolate resembling OTU4227, which lacks niche preferences, for the inoculation experiments. Delving into the functional discrepancies in salt stress adaptation between isolates with and without niche preferences, and comprehending the reasons behind these variations, would greatly enrich the analysis.

Agree. This is a good idea to conduct the functional analysis of isolates with and without niche preferences in salt tolerance and plant growth promotion. In this study, we took this into account and performed a bacterial isolation campaign again, expecting to isolate more native *Pseudomonas* strains. This indeed expanded several *Pseudomonas* species (Supplementary Table 2), however, none isolates closely resembled OTU4227 (the one without niche preferences). We have added sequence similarities between all *Pseudomonas* isolates and OTU4227 in Supplementary Table 2. Although, it should be noted that the relative abundance of OTU2336 (the one with niche preferences) is far higher than the total abundance of OTU4227 in root and rhizosphere, especially in 300 mM NaCl-treated sample (Fig. 3g). Thus, we think isolates resembling OTU2336, like YE17 we selected, are more representative. We have included these considerations in the revised manuscript (Line 205-210), and we have added a note in discussion to highlight the importance of such work (Line 400-402).

Line 205-210: “A total of 34 *Pseudomonas* isolates were obtained (Supplementary Table 2), and their sequences were compared with the two most prevalent sequences, OTU2336 and OTU4227. None of the isolates obtained in this study shared >98% similarity with OTU4227 (Supplementary Table 2). However, one isolate, namely *Pseudomonas frederiksbergensis* YE17, showed 99.21% 16S rRNA gene similarity with OTU2336 (Supplementary Fig. 7 and Supplementary Table 2).”

Line 400-402: “Although we did not get isolates resembling OTU4227, it is important that future studies compare the functions of isolates with and without niche preferences for plant growth promotion.”

Shotgun Metagenomics and NaCl Treatment: The utilization of shotgun metagenomics to assess 300mM NaCl-treated samples, while underscoring the importance of motility, might not be directly comparable to exudates from 100 and 200mM NaCl-treated samples due to disparate conditions. For coherence, it would be judicious to consider measurements of root exudates from 300mM NaCl-treated samples. Additionally, it is

essential for the authors to explicitly state whether the current isolates possess genes pertinent to motility function.

--We appreciated your comments for using consistent NaCl concentration. We indeed treated plant using 300 mM NaCl for root exudates measurement previously. However, all plants in this group died due to excessive NaCl concentration. Thus, this group was out of consideration. Please note, root exudate is usually collected by hydroponics. We have added explanation in the methods (Line 567-570). To avoid confusing, we also provided a detail table listing all samples (including sample number, treatment, and plant culture condition) used for multi-omics (Supplementary Data 3).

Line 567-570: “These seedlings were transferred to the hydroponic system (containing 0, 100, 200 and 300 mM NaCl solution) for root exudates collection. We found all plants in 300 mM solution died due to excessive NaCl concentration, therefore, two salt treatments (100 and 200 mM) were remained.”

--As mentioned above, we have now shown that *Pseudomonas* isolates in this study harbor a series of motility genes, as indicated by the new Supplementary Data 2 and Fig. 12.

Functional Shifts of *Pseudomonas* in MAGs: The presence of four condition-specific MAGs, illustrated in Figure 4g, and derived from co-assembled reads, MAG binning, and de-replication, has raised inquiries regarding the functional shifts of *Pseudomonas*. Given their concurrent enrichment under salt stress (supplementary figure 13), the argument for distinct functional shifts becomes less compelling. If there are *Pseudomonas* that didn't display enrichment under salt stress, a comparison between the four enriched MAGs and those that remained unenriched would be pertinent. If not, it might be advisable to omit this paragraph.

We agree with this reviewer that it is less compelling to compare the functional shift of four MAGs since they all enriched in salt-treated samples. Thus, we decide to remove this paragraph.

Minor Concerns:

Line23: I'd suggest to remove word “metabolites” from the manuscript. We could explore the enrichment pattern only with the microbiome data. Throughout the manuscript, careful revisions should be undertaken to ensure clarity and coherence.

Thanks for your suggestions. We have removed “metabolites”. The current manuscript has been checked carefully.

Precision in *Pseudomonas* Enrichment Assertion: The assertion about salt-induced *Pseudomonas* enrichment in both root and rhizosphere (lines 218-219) necessitates a nuanced approach, considering that not all strains exhibit consistent enrichment patterns across various niches. Ensuring an accurate portrayal of this information is crucial.

Thanks for spotting this. We have modified this sentence.

Line 199-200: “Collectively, these results indicate that salt-induced *Pseudomonas* enrichment occurs in the root and/or rhizosphere soil.”

Enhancement of Image Resolution Quality: Rectifying the issue of low image resolution quality across all figures is vital to guarantee clear and comprehensible visuals.

Thanks for your suggestions. Figures in our original manuscript were compressed when we insert them. We have checked and improved all figures. They are clear now.

Appropriate Use of "Significantly": The term "significantly" (line 106) should be used precisely to reflect the nature of comparisons, taking into account that not all comparisons yield statistical significance.

We have modified this sentence and checked throughout our manuscript to use the term precisely.

Improved Figure Legends: Enhancing figure legends substantially to provide comprehensive descriptions would bolster the reader's comprehension of the visual content. For instance: It's unclear what "EC" represents in figure 1a. Are the labels "0," "100," "200," and "300" indicative of 0mM, 100mM, 200mM, and 300mM in figure

Thanks for your valuable suggestions. We have included more explicit captions for all figures. Abbreviation and labels "0," "100," "200," and "300" have been defined. The number of replicates and statistical analysis have also been added.

Readability of Supplementary Figure 2: Acknowledging the challenges posed by the color scheme in Supplementary Figure 2 and making necessary adjustments to enhance clarity and accessibility is recommended. For instance, addressing the similarity in color between Actinobacteria and Gemmatimonadita would improve differentiation.

We agree with this reviewer and have now changed this figure appropriately. It is Supplementary Figure 3 now.

Clarification of Diversity Drop in Figure 2a: The noticeable decrease in diversity for the 300mM NaCl-treated sample on Day 14 (Figure 2a) in contrast to other salt-treated samples warrants clarification. Providing context and explaining any distinct trends would contribute to a better understanding.

Thank you for this point, we have now edited the manuscript accordingly as follows:

Line 118-121: “The decreased Shannon’s diversity in the 300 mM NaCl treatment in contrast to other salt-treated samples was possibly because the high salt concentration greatly affected the growth of wild soybean, limiting the recruitment of some microbial species.”

Elaboration on Abbreviations: Expanding on abbreviations such as "K," "N," etc. (line 285) would amplify reader comprehension.

We have re-worded this sentence to improve readability and thank the reviewer for pointing this out. Other abbreviations have also been checked.

Line 247-250: “COG analysis revealed that cell motility (COG category: N), transcription (COG category: K), and several biogenesis-, transport-, and metabolism-related processes were markedly enriched in salt-treated rhizosphere soil (Fig. 5a).”

By diligently addressing these major and minor concerns, the manuscript's lucidity, precision, and overall scientific rigor will undergo substantial improvement, leading to a more robust and impactful contribution to the domain of salt stress and plant microbiome interactions.

Thanks for your valuable suggestions and appreciation of our work. We believe that our manuscript has greatly benefited from the constructive feedback.

Reviewer #2 (Remarks to the Author):

This review concerns the manuscript “Purines enrich rhizosphere *Pseudomonas* to improve wild soybean growth under salt stress” by Yanfen Zheng et al.. The authors have studied wild soybean under salt stress and reveal an enrichment of *Pseudomonas* bacteria using cultivation independent methodology. They isolated 30 *Pseudomonas* strains and characterize two of them to enhance plant growth under salt stress. Rhizosphere metagenomics and metatranscriptomics reveal an enhancement bacterial cell motility features in a salty rhizosphere (including a purine-binding chemotaxis protein CheW). A metabolomic analysis reveals then an accumulation of purines in the rhizosphere. The two *Pseudomonas* strains then indeed respond chemotactically towards purines in invitro studies and application of purines enhances plant growth in presence of bacteria. Overall the authors present many interesting and possibly connected findings. However, the major problem is that the causal linkings of these findings were not demonstrated, while the authors report the findings as causally linked and also placing them into too generalized context. For demonstration of causal linkage, often genetic experiments (plant mutant not exuding purines and/or *Pseudomonas* mutants in CheW) proof helpful. Besides this major comment, there are a few further main concerns and numerous minor points that require attention.

We thank this reviewer for the valuable comment, which have helped us to improve our study greatly. We fully agree that genetic experiments are helpful to connect our findings. In fact, it is hard to get wild soybean mutant not exuding purines. Hopefully in the future this could be done. But we performed *cheW* knockout in *Pseudomonas* isolate. Three *cheW* genes were annotated in both two *Pseudomonas* genomes (Supplementary Fig. 12). Two *cheW* mutants (named $\Delta cheW1$ and $\Delta cheW2$) nearly lost or decreased the chemotactic ability to xanthine, emphasizing the importance of CheW in the interaction between *Pseudomonas* and root exudate purine. Furthermore, compared to wild type *Pseudomonas* isolates, the inoculation of *Pseudomonas* mutants ($\Delta cheW1$ and $\Delta cheW2$) lost the plant salt tolerance enhancement. Above results have

been provided in Line 356-367 and Figure 6. We hope you find our revisions satisfactory.

Line 356-367: “To further verify the role of CheW in the *Pseudomonas* chemotaxis and plant salt tolerance enhancement, we deleted three *cheW* genes from *P. stutzeri* XN05-1 and *P. frederiksbergensis* YE17 using the tri-parental conjugation method to obtain the $\Delta cheW1$, $\Delta cheW2$, and $\Delta cheW3$ strains of XN05-1 and YE17. We found that strains XN05-1 $\Delta cheW1$ and YE17 $\Delta cheW1$ lost the chemotactic behavior to xanthine, whereas that in XN05-1 $\Delta cheW2$ and YE17 $\Delta cheW2$ was impaired (Fig. 6e and f). We next used the mutant strains to inoculate wild soybean seedlings under salt stress, and found that $\Delta cheW3$ mutant showed the same extent of plant biomass as the wild-type strains. However, $\Delta cheW1$ and $\Delta cheW2$ mutants of XN05-1 and YE17 both led to a decreased plant biomass compared with the wild-type strains (Fig. 6g). This highlighted the critical role of motility related gene *cheW* in the interaction between the host and root-associated *Pseudomonas*.”

Major comments:

Study context: The title claims that purines enrich *Pseudomonas* bacteria. While this is true for the 2 studied strains, it is not clear if this is a generic *Pseudomonas*- or strain-specific trait. Do other *Pseudomonas* bacteria also enrich on roots after application of xanthine or is this a specific behavior of these two strains? Such a claim requires testing more generally *Pseudomonas* strains and also other bacterial genera.

Very good point that more *Pseudomonas* strains and other genera would also be included. However, the *Pseudomonas* genus now contains more than 300 different species that have been validated (<https://lpsn.dsmz.de/genus/pseudomonas>). We cannot yet properly claim if the *Pseudomonas* enrichment is a strain-specific trait in this study. In fact, not all *Pseudomonas* OTUs in our dataset showed significant enrichment under salt stress (Fig. 3g). We selected isolate that had close genetic similarity to the most abundant *Pseudomonas* OTU from our 16S rRNA gene experiment. To address another comment that if the *Pseudomonas* enrichment on roots after application of xanthine is a genus-specific trait, we conducted a new pot experiment. Xanthine was added into natural soil growing wild soybean seedlings. The rhizosphere and root samples were performed 16S rRNA amplicon sequencing to investigate the enriched taxa. Intriguingly, we found that xanthine application increased *Pseudomonas* abundance significantly, from the relative abundance of 0.81% in control to 17.47% in xanthine treated rhizosphere soil. The fold change and relative abundance of *Pseudomonas* is far higher than other genera (Fig. 6b and c, Supplementary Fig. 17). Therefore, we think the bacterial enrichment induced by xanthine was largely specific to *Pseudomonas* genus. We have provided new figures and description.

Line 316-326: “To further test if the *Pseudomonas* enrichment in the roots after xanthine application was a genus-specific trait, xanthine was added into natural soil growing wild soybean seedlings. The rhizosphere soil and root samples were collected,

and 16S rRNA amplicon sequencing was performed to investigate the enriched microbes by xanthine. We found that xanthine clearly altered the bacterial communities in the root and rhizosphere soil (Fig. 6b). Intriguingly, among the top five most abundant genera, *Pseudomonas* was enriched in the xanthine addition group with highest fold change, which was up to 21.5-fold higher in the xanthine treated rhizosphere soil (relative abundance of 17.47%) than that in the control (0.81%) (Fig. 6c and Supplementary Fig. 17). The results indicate that xanthine-induced bacterial enrichment is largely specific to *Pseudomonas* genus.”

Second, the title also claims that this enrichment leads to growth promotion. Again this has only been shown for the 2 strains without benchmarking other strains for generalization. Furthermore the xanthine-application experiments with the two strains lack an important control: how do the plants grow alone (without *Pseudomonads*!) after application of xanthine. This is necessary to answer if *Pseudomonas* bacteria mediate the growth promotion.

--We appreciate the reviewer's comment regarding the generalization of *Pseudomonas* strains. This study focused on the most abundant *Pseudomonas* OTU relevant strains. The strain generalization will be the good subject of future experiments.

--This is a very interesting and valid question, and the application of xanthine without *Pseudomonas* inoculation has now been carried out. The results showed that xanthine addition slightly, but not significantly, enhanced plant growth, and the promotion effect was further increased in the xanthine inclusion with the presence of *Pseudomonas*. This indicates that the plant growth promotion was mediated by *Pseudomonas* bacteria. We have added this new results in the revised manuscript.

Line 331-338: “To validate if the *Pseudomonas* enrichment correlated directly with the enhanced salt stress tolerance in wild soybean. We performed experiments involving *Pseudomonas* inoculation devoid of xanthine, applying xanthine but lacking *Pseudomonas* and inoculating *Pseudomonas* in the presence of xanthine. The results showed that xanthine addition alone slightly, but not significantly, enhanced plant growth, which was further enhanced in the presence of *Pseudomonas* (Fig. 6d). This result shows that wild soybean growth promotion after xanthine application is dependent on *Pseudomonas*.”

Also the third claim in the title - growth promotion under salt stress - is lacking an important control: The strain inoculation experiments were only performed under salt stress conditions. It remains unclear if the growth promotion is generic or specific to salt stress. Does it occurs also under non-salty conditions.

Great idea to look into this. We have reconducted *Pseudomonas* inoculation experiment under salt and non-salt conditions. The result shows that only one isolate increased the root growth but not shoot growth under non-salt stress (Fig. 4a and c). However, under salt stress, two *Pseudomonas* isolates XN05-1 and YE17 both improved the root and shoot growth (including length and fresh weight) of wild soybean (Fig. 4), suggesting that the growth promotion mediated by *Pseudomonas* isolates is primarily specific to

salt stress. We have added this results in Line 224-229 and prepared a new figure (Fig. 4).

Line 224-229: “We observed that under non-salt stress condition, only strain YE17 increased root growth but not shoot growth (Fig. 4). However, under salt stress, *Pseudomonas* isolates XN05-1 and YE17 significantly improved root and shoot growths (including length and fresh weight) of wild soybean (Fig. 4). This indicates that the plant growth enhancement induced by these strains was primarily specific to salt stress condition.”

Finally, the measurements of rhizosphere chemistry reveal an accumulation of purines in this compartment. Of course it is tempting to assume that they are exuded from plants. Can the authors exclude the possibility that the purine compounds have microbial origin? The work is also placed in the context of the salt tolerant *G. soja* vs. ?salt-sensitive? *G. max*. The study covers data from both species, but rarely as a direct comparisons. E.g in Fig. 1 it would be nice to document the phenotype of *G. max*. Alternatively, the context of differential salt tolerance by these plants is not necessary.

--Root exudates were collected from seedlings growing in sterile hydroponic system. Thus, the possibility that the purine compounds derived from microorganisms could be excluded. To be clearer, we have detailed methods and clarified this.

Line 568-570: “Since the root exudates were collected from seedlings growing in sterile hydroponic system, the possibility that compounds derived from microorganisms could be excluded.”

--The different salt tolerance between *G. soja* and *G. max* was not a major concern in this study. For coherence, we have decided to remove the metabolic data of *G. max*, focusing on *G. soja* in the revised manuscript.

Data connectivity: The connectivity of the different data often appears a bit occasional and should be improved. It would be important to better link the OTU data (representative OTU sequence) with the strains (full length 16S sequence (SANGER?), their genome) and the MAGs of the metagenomic sequencing. These different data share the same taxonomy assignment (Genus=*Pseudomonas*), but it is missed opportunity to compare the actual DNA sequences detected by the different approaches. This can be done simply by alignments and quantifying % sequence similarity. It is unclear if the isolated strains belong to the abundant MAGs. The credibility (how well the 2 strains can represent the culture independent findings) of data connection would greatly increase if these two isolates would fully correspond to the sequencings. A few further things appear a bit occasional such as why presenting the top 10 *Pseudomonas* (Fig. S7) compared to top 12 genera (Fig. 3).

--Very good point. This indeed our selection strategy for representative *Pseudomonas* strains. Now, we have provided a figure of DNA sequence alignments (Supplementary Fig. 7). The 16S rRNA gene sequence of strain YE17 showed 99.21% match to the salt-induced OTU2336 (the most abundant *Pseudomonas* OTU). We also added this information to the revised manuscript (Line 208-210). Additionally, description of

MAGs has been removed as we and reviewer #1 think it is less compelling to show their functional shift. To show whether another strain (XN05-1) represented the actual DNA sequences, its genome sequence was mapped to metagenomic reads. We have added this analysis in Line 214-217.

Line 208-210: “However, one isolate, namely *Pseudomonas frederiksbergensis* YE17, showed 99.21% 16S rRNA gene similarity with OTU2336 (Supplementary Fig. 7 and Supplementary Table 2).”

Line 214-217: “The number of metagenomic reads mapped to the XN05-1 genome was higher in the salt treatments than in the control (Supplementary Fig. 8b), suggesting that this strain represents the actual DNA sequences enriched in salt treatments.”

--We have modified Fig. 3 to show the top 10 genera.

RNA seq analysis: the fact that 77/189k genes vary in response to salt suggests a massive transcriptional reprogramming and then the following sentence mentions only fold changes of 1.4x and 1.6x (not reflecting a massive transcript response). I can hardly believe that so many genes vary with such a low fold change. I wonder about the number of replicates and the statistic threshold for this claim. FDR correction of p-values?

--We recognize this may be confusing, but we rechecked our data and found this statement was true. The fold changes of 1.6 or 1.4 were calculated based on the expression levels of all genes annotated as COG N or J, including up-regulated, down-regulated and non-differential genes. If we only consider the expression levels of all differentially expressed genes (DEGs), the fold change of COG N will be 2.5. In fact, each COG category contains numerous genes, therefore, function alteration with large fold change at this COG level is not too common. This analysis just clued us to find salt-responsive genes at the finer level, like *cheW* and *fliC* in this study. To avoid misunderstanding, we have reformatted this claim.

Line 255-260: “After illumina sequencing, we obtained a total of 189,319 genes in our dataset, of which, 96,803 genes were identified as the differentially expressed genes (DEGs). COG function analysis revealed that DEGs associated with cell motility (COG category: N) and translation, ribosomal structure, and biogenesis (COG category: J) were enriched 2.5- and 2.0-folds enriched in salt treatment, respectively (Fig. 5b).”

--In this study, three replicates were collected for metatranscriptome sequencing due to the expensive cost and large sample amount required to get high-quality RNA (*n* has been shown in Fig. 5b). In the data analysis, the cutoff of DEGs was set as fold change $> = 2$, and FDR adjusted $P < = 0.05$. We have added relevant description to methods and the legend of Figure 5.

Line 541-542: “For metatranscriptomic sequencing, independent rhizosphere samples ($n = 3$ biological replicate) were prepared according to above description.”

Line 559-561: “EdgeR (Empirical analysis of Digital Gene Expression in R) was applied to identify DEGs between control and salt treatment, with the cutoff of fold

change ≥ 2 , and false discovery rate (FDR) adjusted $P \leq 0.05$.”

Metabolomics: Several aspects of the metabolomic analyses require clarification. To which type of *Pseudomonas* data (amplicon-genus, amplicon-OTU, Metagenome...) were they correlated to? The massive fold changes (\log_{10} !) suggest that these compounds were hardly measured in the control samples. Pls clarify coverage of control vs. treated samples and their normalization. Similar to OTU, metagenome or RNAseq data, also the metabolite data requires dual information on abundance and fold change for good interpretation. The authors have chosen dot plots in Fig. 3 for such dual information of the OTU data. Would it be possible to apply this principle also to the metabolite data? E.g. using MA plots?

--Apologized for this unclear. The *Pseudomonas* for correlating to metabolomic data was the abundance of *Pseudomonas* genus from 16S rRNA gene amplicon data. We have clarified this in the text and figure legends.

Line 298-299: “Metabolites correlated with *Pseudomonas* genus from the amplicon data were investigated,”.

--Indeed, the contents of compounds we mentioned here were hardly measured in the control group, suggesting they were only secreted by plant under salt stimulation. The detailed values of compounds in control and salt-treated samples were shown in Supplementary Fig. 15, and the relevant statement was added in the result (Line 305-308). Additionally, we have provided additional description on data normalization in the method (Line 578-581).

Line 305-308: “These compounds were hardly measured in the control but were secreted by salt-treated plants in large quantities (Supplementary Fig. 15), suggesting that they were specific compounds correlated with salt stress.”

Line 578-581: “Metabolites were identified using both mass spectrum and retention time. To compare the content of each metabolite between the control and treatments, the mass spectrum peak of each metabolite in different samples was calibrated according to integration of the peak area.”

--We agree that dual information on abundance and fold change is better to exhibit our data. We have added abundance analysis for metagenome and metatranscriptomic data (Fig. 5a and b). As metabolites correlating to *Pseudomonas* were too many, it is hard to display using dot plot. We have therefore maintained circle heatmap and circle bar plot to show abundance and fold change, respectively (Fig. 6a).

Language: The manuscript requires editing by a native speaker. There are many ‘linguistic bumps’. Furthermore some terminology needs to be clarified: is microbiota synonymous for microbiome? Difference between resistance and tolerance to salt stress? Consistent use of hyphen: root-associated, salt-induced... Attention to vocabulary, too: do not use terms without “direction”. Example in L147: Instead of describing “shifts”, specify whether you refer to and increase or decrease.

--Thank you for this valuable comment. The manuscript has been edited thoroughly by a native speaker.

--The terms “microbiota” and “microbiome” are often interchangeable. But there are some subtle differences between the two terminologies. We have clarified them (Line 54-56) and checked their usage in our revised manuscript.

Line 54-56: “Microbiota refers to the microbial community in a particular environment, while the microbiome comprises microbiota and their structural elements (such as nucleic acids and proteins) and metabolites¹¹.”

--Resistance is used to describe the inherited ability of plant to grow under biotic or abiotic stresses, attributing to the genetic factor. Tolerance is a plant’s ability to survive under biotic or abiotic stresses, whether inherited or not, typically attributing to plant vigor and regrowth of damaged tissue. Based on this definition, we think “tolerance” is more accurate in our text. We have checked and modified.

--The problems of hyphen and vocabulary have been corrected.

Figure legends are often insufficient or incomplete: A figure together with its caption should be self-explanatory. Detail the experiment and method with one sentence, the measurements must be explained, abbreviations defined, number of replicates ($n = X$) is missing throughout the manuscript. A caption shall also state the statistic test and eventual correction of P-values for multiple hypothesis testing. Figure 1 comes with a particularly rudimentary caption, no panel indexes in the title.

Many thanks for your suggestions. We have included more explicit captions for all figures.

Figure 3: The relative abundance plot in 3a-c is misleading. As it is it displays cumulative relative abundance of the 3 treatments. The 3 treatments should be shown side-by-side. Why not a dot plot with 4 columns (control, 100-300) where the dot size corresponds to relative abundance? To make the claim that root changes are strongest, it would be helpful to use the same scale for root, rhizosphere and soil. Since the work with the 2 *Pseudomonas* isolates is described in a separate paragraph, consider a separate main figure. Use boxplots instead of these non-scientific violine plots (individual replicates are poorly resolved).

--We appreciated your valuable suggestions. We have redrawn Figure 3. Please note that dot size cannot correspond to relative abundance as dot plot described the fold change of 3 salt treatments compared to control, which resulted in 3 dots. In other words, if dot size corresponds to relative abundance, the relative abundance of control group will be missing. Therefore, additional bar plot is used to show relative abundance of 4 groups side by side with the same scale for root, rhizosphere soil, and bulk soil. Please see Figure 3a-c.

--The work of two *Pseudomonas* isolates inoculation has been separated from Figure 3. It is Figure 4 now.

Minor comments

L43: *many* strategies, and then only 2 examples follow

We have added one additional example.

Line 40-43: “Salt-tolerant plants have evolved many strategies to resist salt stress, including adjusting cellular osmotic pressure by biosynthesis of osmoprotectants¹, secreting salt out of plants via glands or trichomes², and maintaining cellular redox equilibrium³.”

L47: what are “halophyte root associated environments”?

We have changed this sentence to “Various beneficial microbes reside in root-associated environments of halophytes⁵⁻⁸, ...”

L50: Bacterial provision of nutrients should alleviate salt stress. It is hard to believe that this is the most important physiological process.

Sorry for this unclear. It is indeed one of ways (like phosphate solubilization and nitrogen fixation) but not the most important way for salt stress alleviation. Therefore, this sentence has been adjusted.

Line 47-49: “They can alleviate the adverse effects of salt stress on plant growth through various physiological regulatory processes, such as maintaining ion homeostasis and altering endogenous hormone status¹⁰.”

L75-78: rephrase

This sentence has been modified.

Line 77-79: “Previous studies observed that salt-stressed plants secreted larger amounts of metabolites, including phenolic compounds, amino acids, organic acids, and sugars, than non-stressed plants^{32,33}.”

L99: EC based on available ions, no?

We have modified it.

Line 97-99: “To determine the salt stress levels, we measured the soil electrical conductivity (EC), which reflects the available ion content in the soil.”

L101: precise that TC and TN are soil measures

We have explicitly stated them. “the content of **soil** total carbon (TC) and total nitrogen (TN)”.

Inconsistent usage salt treatments in text and figures. Sometimes used as control/S100/S200/S300 or 0/100/200/300

Thanks for your suggestion. They are consistent throughout text and figures now.

L122/Fig2a: root 300 is not decreased at 14d

We have made adjustment.

Line 116-118: "...whereas all salt treatments (except for the 300 mM NaCl treatment) led to a significantly increased level of Shannon's diversity at Day 7 and Day 14."

L125: refer to NMDS stress values.

Thanks for your suggestions. To better present our data, NMDS was replaced by PCoA with ANOSIM values (Fig. 2b) and additional PCoA performed on plant compartment at Day 1, Day 7 and Day 14 was provided in Supplementary Fig. 1. The relevant description has been modified.

Line 126-131: "The root community (analysis of similarity [ANOSIM]; $R = 0.76$, $P < 0.01$; at Day 14) showed greater compositional changes after salt stress than the other two compartments (ANOSIM; rhizosphere soil: $R = 0.41$, $P = 0.002$; bulk soil: $R = 0.27$, $P = 0.023$; at Day 14), which were also revealed by a larger separation between the control and salt treatments in the root samples (Fig. 2b and Supplementary Fig. 1)."

L125/L128: demonstrate and indicate seem contradicting verbs for describing the same finding

We have reformatted these two sentences (removed "demonstrate") as follow:

Line 129-132: "..., which were also revealed by a larger separation between the control and salt treatments in the root samples (Fig. 2b and Supplementary Fig. 1). These results indicate that salt stress exhibits a greater effect on root communities."

L134...: The claims of that paragraph are not sustained with statistics in Figure S1. Is it useful at all to mention a decrease of Proteobacteria, when the next sentence then reports increase of Pseudomonadales and Xanthomonadales? High rank descriptions (Phylum or Order level) are often not very informative. I find the genus description sufficient.

Thanks for your suggestions. We have changed Figure S1 (Proteobacteria was shown at class level) and reduced high rank descriptions.

L146: Why numerically mentioning the decrease of Rhizobiales, while not for the others? The story is about Pseudomonadales.

We have modified this sentence and removed the description of Rhizobiales.

L151: it is unclear what these percent values (e.g. 21.74% of root genera) refer to. I cannot find this in Fig. S2 or Table S2. Cumulative relative abundance, fraction of genera?

We apologise for our unclear description. These percent values are the proportions of the number of genera with significant change to all genera in root, rhizosphere soil, and bulk soil. We have added detail description in the text.

Line 147-150: "After 14 days of salt stress, 130 of 598 genera (21.74%) in the root, 56 of 776 genera (7.22%) in the rhizosphere soil, and 38 of 779 genera (4.88%) in the bulk

soil showed significant enrichment or depletion in at least one salt treatment (Supplementary Fig. 3 and Supplementary Data 1).”

L161: avoid such suggestive interpretation
We have deleted the offending sentence.

L163: Enrichment of Proteobacteria? Inconsistent with L141
Proteobacteria has been shown at the class level. Gammaproteobacteria was enriched with high relative abundance.

Fig. 2: endosphere or root?
We have changed to “root” and checked throughout figures and text.

L189: cite first Fig. 3a then 3b then 3c
We cited first Fig. 3a and 3b. They were in the previous line of 3c.

L206: to discussion
We changed this sentence here and discussed this result in discussion section.
Line 184-187: “The results showed that the relative abundance of *Pseudomonas* was higher under salt stress than that in control in all cases (Supplementary Fig. 6), indicating that salt-induced *Pseudomonas* enrichment was conserved across different microbiome backgrounds.”
Line 392-395: “The presence of *Pseudomonas* across multiple plants indicated this genus might establish mutualistic interaction with their hosts³⁵. It is important to evaluate if this close association is consistent across a broader range of host plants and soil types.”

L202: geographic distance is a poor discrimination for different soils. Use soil type definitions and/or chemical analyses instead.
Agree. We have provided soil properties.
Line 182-184: “..., another Shandong soil (collected from ~ 400 km away from the soil used in this study; pH: 8.8 ± 0.2 , EC: 175.3 ± 30.2 $\mu\text{s}/\text{cm}$; Supplementary Fig. 5).”

L224: isolation from roots of salt stressed plants?
Indeed. We have added this information.

L237: show data of 'before' and 'after' salt stress
We have reperformed this experiment and changed text according to latest results and methods.

L243: equal culture?
We have changed to “equal amounts”.

L243: qPCR is a quantitative and not an observatory method

Thanks for your comments. We have removed “observe” and modified this sentence.
Line 230-233: “To confirm that *Pseudomonas* isolates colonized the roots of wild soybean, we inoculated these two strains in sterile soil in equal amounts to quantify the levels of *Pseudomonas* colonization under salt and non-salt conditions using qPCR with specific primers.”

L249: The genome sequencing cannot be claimed to determine the genetic basis. It is simply descriptive for what genes these two strains have. There is not determination whether IAA, ACC... are involved here. How does the 16S sequence of the genome compare to the Sangersequence of the strain, the OTU and the MAG data?

Thanks for your suggestions. This part has been removed. We have provided the information of motility-related genes in *Pseudomonas*. Additionally, sequence alignment of strain and OTU, as well as strain and metagenomic reads have been provided.

L265/344: bule

Sorry for this mistake. We have changed to “blue”.

L312: expressing, not encoding

We have changed it.

Fig. 4g: I don’t get the color link between MAGID and Treatment in the legend.

Fig. 4g and its description in the text have been removed.

L322: which groups?

This part has been deleted.

L354: why two and not three salt treatments?

We indeed performed three salt treatments. Please note, root exudate is usually collected by hydroponics. All plants in 300 mM NaCl treatment died. Thus, this group was out of consideration. We have added explanation in the methods. To avoid confusing, we have also provided a detail table listing all samples (including sample number, treatment, and plant culture condition) used for multi-omics (Supplementary Data 3).

Fig. 5a: would be better to display the fold changes as being an enrichment (+) or a depleting (-) and not both on same axis.

Thanks for your suggestions. The fold changes of enrichment and depletion were shown by outward and inward bars, respectively. To avoid confusions, different colors have been labeled.

Figures 5d/e require important controls: how do the plants grow alone with application of xanthine (without *Pseudomonads*).

Agree. Thanks for your valuable suggestions. As mentioned above, we have now shown the xanthine control.

L447: I do not understand why these traits are discussed here. They do not link to the data of this study.

We have removed these sentences.

L479: The experiment had a time frame of 2 weeks – is this sufficient that HGT could explain genome changes? Isn't it more likely that different strains become more abundant and this is reflected by the metagenomic analysis?

Agree. We have deleted the description of HGT and functional changes of MAGs in the manuscript.

L557: are the four “biological” replicates pooled samples of six pots/plants. Are these 1 plant per pot or 6 (L563)?

We apologise that this was unclear. Six plants per pot generated one replicate. We have added this statement in the manuscript, as below:

Line 485-486: “The experiment was performed using four biological replicates (six wild soybean plants in each pot generated one replicate) for all treatments.”

Reviewers' Comments:

Reviewer #1:

Remarks to the Author:

I express my gratitude to the authors for thoroughly revising the study. I appreciate that my concerns regarding result generalization have been adequately addressed. Nonetheless, I have identified some errors in the manuscript, such as on Line 242, where it should state "three salt treatments" instead of two. I kindly request the authors to review the entire manuscript carefully.

Reviewer #2:

Remarks to the Author:

This review concerns the revised manuscript "Purines enrich root-associated *Pseudomonas* to improve wild soybean growth under salt stress" by Yanfen Zheng et al..

The authors have well revised their work with incorporating the suggestions of both referees. Particularly, I think that generation and testing of the *Pseudomonas* mutants was instrumental to actually document that purines function in enrichment of the two *Pseudomonas* strains, which in turn promote soybean growth under salt stress. Also the newly added experiments with the key controls were important to convincingly show that the xanthine~*Pseudomonas* observation is salt-stress specific.

While, the manuscript has greatly improved and gained in clarity, I have a concern regarding the new Fig. 6ef: how should this assay show chemotaxis? It is unclear how the 'quantitative soft agar plate assay' should allow to assess chemotaxis. What I read from the methods, is that the strains are injected into the agar at equal ODs and that agar contains 1 mM xanthine. How can this assay show 'growth towards xanthine' when xanthine is uniformly distributed in the agar plate? It is unclear why the authors did not use their 1-ml-syringe-needle assay, which truly shows chemotaxis.

Otherwise, I only have some minor suggestions for further improvement:

- Microbiota: Similar to microbiome/s, the term microbiota is singular and refers to one microbial community. Edits to the abstract are necessary: "The root-associated microbiota plays..."; "...that a highly conserved microbiota...".
- Rephrase L24: Change "...salt-stressed root and rhizosphere soil..." to "...root and rhizosphere microbiota of salt-stressed soybean plants...".
- L25: The salt tolerance of wild soybean should be mentioned earlier.
- Revise L29: Don't you want to say that "...roots of salt stressed plants secreted purines, especially xanthine, which induced motility of the *Pseudomonas* isolates. Moreover, exogenous application of xanthine to non-stressed..."?
- Rephrase L31: "...*Pseudomonas* mutant analysis showed that...and for enhancing plant salt tolerance."
- Fig. 6c: I would use the same scale for both plots as this shows more directly the root-enrichment compared to rhizosphere.
- In line 359, the generation of cheW mutants in the 3 genes in both strains is mentioned, but their chemotactic behavior is only reported for the cheW1 and cheW2 strains in the text, while Fig. 6ef shows also the cheW3 mutants. I would mention this, as this is consistent with the cheW3 mutant strains not promoting soybean growth under salt stress.
- Rephrase L363: "a bacterial mutant cannot show plant biomass"

Response to the editor and reviewers (NCOMMS-23-25309B)

Many thanks to the editor and reviewers for their constructive and helpful comments. We have thoroughly revised the manuscript in response to these comments (changes in the revised manuscript are in red). We greatly appreciate the time that went into providing such detailed reviews and hope we have addressed the concerns sufficiently. We now present our point-by-point responses to the reviewers:

Reviewer #1 (Response):

I express my gratitude to the authors for thoroughly revising the study. I appreciate that my concerns regarding result generalization have been adequately addressed. Nonetheless, I have identified some errors in the manuscript, such as on Line 242, where it should state "three salt treatments" instead of two. I kindly request the authors to review the entire manuscript carefully.

We thank the reviewer for the comment and are pleased that our edits addressed their concerns.

Sorry for this mistake. We have carefully checked our manuscript.

Reviewer #2 (Response):

This review concerns the revised manuscript "Purines enrich root-associated *Pseudomonas* to improve wild soybean growth under salt stress" by Yanfen Zheng et al..

The authors have well revised their work with incorporating the suggestions of both referees. Particularly, I think that generation and testing of the *Pseudomonas* mutants was instrumental to actually document that purines function in enrichment of the two *Pseudomonas* strains, which in turn promote soybean growth under salt stress. Also the newly added experiments with the key controls were important to convincingly show that the xanthine~*Pseudomonas* observation is salt-stress specific.

We thank the reviewer for their positive comments, and are pleased that they feel the manuscript has been improved. Below we address the minor things further.

While, the manuscript has greatly improved and gained in clarity, I have a concern regarding the new Fig. 6ef: how should this assay show chemotaxis? It is unclear how the 'quantitative soft agar plate assay' should allow to assess chemotaxis. What I read from the methods, is that the strains are injected into the agar at equal ODs and that agar contains 1 mM xanthine. How can this assay show 'growth towards xanthine' when xanthine is uniformly distributed in the agar plate? It is unclear why the authors did not use their 1-ml-syringe-needle assay, which truly shows chemotaxis.

We thank this reviewer for the question. The soft agar plate assay is quite useful for screening for chemotaxis system gene mutants (Sampedro et al., 2015; Mo et al., 2022;

Jiang et al., 2016). As shown in the schematic diagram (see below), this method is based on a chemical gradient created by the bacterial consumption for the attractant (e.g. xanthine in this study), which can activate the chemotactic response (Sampedro et al., 2015; Keegstra et al., 2022). The bigger expanding ring, the stronger chemotaxis ability. We selected this assay as it could compare the chemotactic behaviors of wild-type and mutant strains visually. These explanations have been added in the revised manuscript (Line 677-680).

Line 677-680: “This method is based on a chemical gradient created by the bacterial consumption for the attractant, which can activate the chemotactic response⁷⁷. We selected this assay here as it could compare the chemotactic behaviors of wild-type and mutant strains visually.”

The schematic diagram of soft agar plate assay for chemoattraction.

References:

- Jiang, N. *et al.* A chemotaxis receptor modulates nodulation during the *Azorhizobium caulinodans*-*Sesbania rostrata* symbiosis. *Appl Environ Microbiol.* **82**, 3174-3184 (2016).
- Keegstra, J. M., Carrara, F. & Stocker, R. The ecological roles of bacterial chemotaxis. *Nat Rev Microbiol.* **20**, 491-504 (2022).
- Mo, R., Ma, W., Zhou, W. & Gao, B. Polar localization of CheO under hypoxia promotes *Campylobacter jejuni* chemotactic behavior within host. *PLoS Pathog.* **18**, e1010953 (2022).
- Sampedro, I., Parales, R. E., Krell, T. & Hill, J. E. *Pseudomonas* chemotaxis. *FEMS Microbiol. Rev.* **39**, 17-46(2015).

Otherwise, I only have some minor suggestions for further improvement:

-Microbiota: Similar to microbiome/s, the term microbiota is singular and refers to one microbial community. Edits to the abstract are necessary: “The root-associated microbiota plays...”; “...that a highly conserved microbiota...”.

We have edited them as suggested.

-Rephrase L24: Change “...salt-stressed root and rhizosphere soil...” to “...root and rhizosphere microbiota of salt-stressed soybean plants...”.

We have edited this sentence as suggested.

-L25: The salt tolerance of white soybean should be mentioned earlier.

We have added relevant description.

Line 22-23: “Here, by focusing on a salt-tolerant plant wild soybean (*Glycine soja*), we demonstrate...”

-Revise L29: Don’t you want to say that “...roots of salt stressed plants secreted purines, especially xanthine, which induced motility of the *Pseudomonas* isolates. Moreover, exogenous application fo xanthine to non-stressed...”?

We have revised this sentence as below:

Line 28-31: “We further found that roots of salt stressed plants secreted purine, especially xanthine, which induced motility of the *Pseudomonas* isolates. Moreover, exogenous application for xanthine to non-stressed plants resulted in *Pseudomonas* enrichment...”

-Rephrase L31: “...*Pseudomonas* mutant analysis showed that...and for enhancing plant salt tolerance .”

We have edited this sentence as suggested.

-Fig. 6c: I would use the same scale for both plots as this shows more directly the root-enrichment compared to rhizosphere.

We have changed to the same scale.

-In line 359, the generation of *cheW* mutants in the 3 genes in both strains is mentioned, but their chemotactic behavior is only reported for the *cheW1* and *cheW2* strains in the text, while Fig. 6ef shows also the *cheW3* mutants. I would mention this, as this is consistent with the *cheW3* mutant strains not promoting soybean growth under salt stress.

We have mentioned the *cheW3* mutant in Line 361-362.

Line 361-362: “We found that the strains XN05-1 Δ *cheW3* and YE17 Δ *cheW3* still showed obvious chemotaxis toward xanthine.”

-Rephrase L363: “a bacterial mutant cannot show plant biomass”

We have edited this sentence.

Line 364-368: “We next used the mutant strains to inoculate wild soybean seedlings under salt stress, and found that Δ *cheW3* mutant, the one still having chemotaxis ability, displayed the same extent of plant biomass as the wild-type strains. However, Δ *cheW1* and Δ *cheW2* mutants of XN05-1 and YE17 both cannot show plant biomass enhancement (Fig. 6g).”